# Lost in Aggregation

## On a Fundamental Expressivity Limit of Message-Passing Graph Neural Networks

## Abstract

We define a generic property for aggregation functions, capturing a vast range of practical aggregations, and prove that any Message-Passing Graph Neural Network (MP-GNN) model with such aggregations induces only a polynomial number of equivalence classes on all graphs - while the number of non-isomorphic graphs is super-exponential (in number of vertices). Adding a familiar perspective, we observe that merely 2-iterations of Color Refinement (CR) induce at least an exponential number of equivalence classes, making the aforementioned MP-GNNs relatively infinitely weaker.

Previous studies state that sum-aggregation MP-GNNs match full CR however they consider a weak, 'non-uniform', notion of distinguishing-power where each graph size may require a different MP-GNN to distinguish graphs up to that size.

Our results concern both distinguishing between non-equivariant vertices and distinguishing between non-isomorphic graphs.

## 1 Introduction

Message-Passing Graph Neural Networks (MP-GNNs) (Kipf & Welling, 2017; Gilmer et al., 2017) are a class of parameterized algorithms for graphs, often used as architectures in graph learning tasks. Such tasks may be learning on graphs that represent molecules and biological structures (Gilmer et al., 2017; Gaudelet et al., 2021), graphs that represent social networks and knowledge bases (Yasunaga et al., 2021), and graphs that represent combinatorial-optimization problems (Tönshoff & Grohe, 2025; Tönshoff et al., 2023). Hence, characterizing the expressivity of MP-GNNs is of great importance.

An MP-GNN is defined by a sequence of layers $L_1, \ldots, L_m$ for some $m \in \mathbb{N}$, each layer $L_t = (\mathsf{mlp}_t, \mathsf{agg}_t, \mathsf{msg}_t)$ comprising a message; aggregation; and combination functions. The combination is implemented always by a Multilayer Perceptron (MLP), and in this paper all MLPs are ReLU-activated and rationally-weighted. Denote by $N_G(v)$ and $v^{(0)}$ the neighborhood and initial feature of a vertex $v$ in a graph $G$, respectively, then $v$'s value after applying layer $t + 1$ is

$$v^{(t+1)} := \mathsf{mlp}_{t+1}(v^{(t)}, \mathsf{agg}_{t+1}\{\!\{\, \mathsf{msg}_{t+1}(v^{(t)}, w^{(t)}) | w \in N_G(v)\}\!\})$$

That is, the layers are applied sequentially, each layer applied in parallel to all vertices: Computing a message for each neighbor; aggregating the messages; and combining the aggregation value with the subject-vertex value. Note that the aggregation can be any function on multisets, with a fixed output-dimension, and in this paper a computable one. For graph-level tasks, an MP-GNN model

has a final *readout* step $R = (\mathsf{mlp}_R, \mathrm{agg}_R)$ comprising an aggregation of the final vertices' values followed by the operation of a final MLP. Denote the readout value for a graph $G$ by $G^{(R)}$, then

$$G^{(R)} := \mathsf{mlp}_R(\mathrm{agg}_R\{\!\{v^{(m)} : v \in V(G)\}\!\})$$

The MLP part of the layers gives MP-GNNs their learnability qualities. The node-level definition of the algorithm, together with the fixed-dimension output aggregation, mean every GNN model can technically be applied to graphs of all sizes and degrees. Finally, no order or unique-ids of the nodes are considered, only the nodes' features and graph structure, hence GNNs are invariant to isomorphism.

A necessary condition for an MP-GNN model to *express* a function, i.e. approximate it by some $\varepsilon$, is to have the adequate distinguishing-power i.e. to output different values for every two inputs on which the function differs (by $> 2\varepsilon$). Thus, we are interested in the distinguishing-power of MP-GNN architectures. A well-studied algorithm for *distinguishing* vertices and graphs is the Color Refinement (CR) algorithm (a.k.a. Weisfeiler-leman algorithm (Morgan, 1965; Weisfeiler & Leman, 1968), see also (Cardon & Crochemore, 1982; Paige & Tarjan, 1987; Berkholz et al., 2017; Grohe, 2021)): An iterative local algorithm which assigns a *color* to each node. In each iteration, the color of each node is updated by adding to it the multiset of its neighbors' current colors. Given a graph $G$, CR runs for $|G|$ iterations by which point maximum granularity of color-classes is reached. The color of a graph after each iteration is the multiset of current colors of its vertices. For $t \in \mathbb{N}$ we denote the algorithm that runs the first $t$ iterations of CR by $\mathrm{CR}^{(t)}$.

It is known that the distinguishing-power of MP-GNNs is upper-bounded by that of CR (Xu et al., 2019; Morris et al., 2019; Aamand et al., 2022). It has also been shown there that the CR bound is tight i.e. there exists an MP-GNN model that distinguishes graphs and vertices if they are distinguishable by CR, however the proof is in a *non-uniform* notion: It proves existence of a distinguishing model **per graph size**. That setting has limited relevance to practice as it implies that a learned model can be correct only on graphs of sizes up to the maximum training-graph size. Such model will be incorrect in many practical scenarios: When there are not enough resources to train on large graphs or when the graphs grow over time.

The notion by which it is required to have (at least) one model that is correct on graphs of all sizes is called *uniform*, and this is the notion of distinguishing-power and expressivity that we consider in this paper. There, the following are straightforward:

1. MP-GNNs do not subsume the distinguishing-power of CR, if only because the value assigned to a vertex by an MP-GNN with $m$ layers is not affected by nodes in distance $> m$.

2. With an auxiliary-dimension initialized to '1', a trivial sum-aggregation MP-GNN subsumes $\mathrm{CR}^{(1)}$, as the sum of that dimension amounts to the number of neighbors.

3. With no restriction on the aggregation function other than being computable and having a fixed output dimension, MP-GNNs with $m$ layers subsume the distinguishing-power of $\mathrm{CR}^{(m)}$ by having an aggregation that simply implements CR and encodes the state in one rational number. However, the use of such information for an MLP, in expressing a target function, is limited i.e. such aggregation is less relevant to practice.

The above calls for a general characterization of practical aggregations, and for bounding[1] their distinguishing-power by a range narrower than $[\text{CR}^{(1)}, \text{CR}]$.

Upper bounds that relate directly to function approximation are proved in several works: In terms of logic (Barceló et al., 2020); circuit complexity (Grohe, 2023); or comparative between different MP-GNNs sub-classes (Rosenbluth et al., 2023; Grohe & Rosenbluth, 2024). In all these however, excluding to some extent (Rosenbluth et al., 2023, Section 6), only specific aggregations are considered.

In (Corso et al., 2020) an inexpressivity result for a general class of aggregations is given, however it is proved only for one message-pass iteration; it assumes that the feature-domain is the real numbers - not only finite precision; and it assumes that the aggregation function is continuous. The domain assumption is unnecessarily permissive - with respect to practice - as operations on infinite-precision real numbers are incomputable, and the assumption on the aggregation functions is unnecessarily restrictive as computable functions can be non-continuous.

In (Khalife & Basu, 2023) it is essentially shown that with exponential activation functions, such as *sigmoid;tanh*, the distinguishing-power of MP-GNNs subsumes $\text{CR}^{(2)}$. However, these functions cannot be precisely computed, hence the result does not apply to computable MP-GNNs. See Section 4 (future research (3)) for further discussion.

Recently, a tight bound has been shown (Rosenbluth & Grohe, 2025) both for the distinguishing-power and the expressivity of *recurrent* MP-GNNs (going back to (Scarselli et al., 2008; Gallicchio & Micheli, 2010)), highlighting the missing knowledge about (non-rec.) MP-GNNs even further.

**New Results**

We consider the domain of graphs with boolean-features vertices, which represents all domains with features over a finite set of finite-precision values. We describe a general class of aggregation functions (Definition 3.1) which captures most of the reasonable aggregations that do not involve exponentiation or division by a graph-size-dependent value, and we analyze their effect on the distinguishing-power of MP-GNNs.

Denote by $\mathcal{N}$ the class of MP-GNNs comprising (only) such aggregations, denote the number of equivalence classes that an MP-GNN $N$ induces on vertices in graphs of size $n$, and on whole graphs of size $n$, by $N_{\text{dp}}(n)$ and $N_{\text{gdp}}(n)$ respectively, and similarly for $\text{CR}^{(2)}$ by $\text{CR}^{(2)}_{\text{dp}}(n)$ and $\text{CR}^{(2)}_{\text{gdp}}(n)$, then we prove the following.

1. The uniform distinguishing-power of each MP-GNN in $\mathcal{N}$ is at most polynomial in the graph size (Theorem 3.8). Formally,

$$\forall N \in \mathcal{N} \ N_{\text{dp}}(n) = \text{poly}(n), \ N_{\text{gdp}}(n) = \text{poly}(n)$$

   As the number of non-isomorphic graphs is super-exponential, $2^{\Omega(n^2 - n \log n)}$, that bound is significant.

2. Observing a lower-bound for $\text{CR}^{(2)}_{\text{dp}}$, we add that not only the distinguishing-power of $\mathcal{N}$ is weaker, i.e. for every $N \in \mathcal{N}$ there are vertices distinguishable by $\text{CR}^{(2)}$ and not by any $N$,

---

[1]To be precise, by referring to $\text{CR}^{(t)}$ as a strict bound we do not imply inclusion but rather that it is not subsumed by MP-GNNs. Obviously, when considering graphs of diameters larger than $t$, there are nodes distinguishable by a trivial MP-GNN with $t + 1$ layers and not by $\text{CR}^{(t)}$.

but it gets infinitely weaker as the graph size grows (Corollary 3.10). Formally,

$$\forall N \in \mathcal{N} \ \lim_{n \to \infty} \frac{N_{\mathrm{dp}}(n)}{\mathrm{CR}_{\mathrm{dp}}^{(2)}(n)} = 0, \ \lim_{n \to \infty} \frac{N_{\mathrm{gdp}}(n)}{\mathrm{CR}_{\mathrm{gdp}}^{(2)}(n)} = 0$$

While we focus on MP-GNNs that consist of ReLU-activated MLPs for their message and combination functions, our results may apply also to other MP-GNNs architectures (Remark 3.7).

## 2  Preliminaries

By $\mathbb{N}; \mathbb{Z}; \mathbb{Q}$ we denote the natural, integer, and rational numbers respectively. For $m \in \mathbb{N}$ we define $[m] \coloneqq \{i : i \in \mathbb{N}, 1 \le i \le m\}$. For a set $S$ and size $m \in \mathbb{N}$ we denote the set of all multisets of size $m$ with elements from $S$ by $\left(\!\binom{S}{m}\!\right)$, and of any finite size by $\left(\!\binom{S}{*}\!\right)$. Let $X$ be a multiset of rationals or a multiset of rational vectors, we define $\mathrm{cd}(X)$ to be the *least common denominator* of the elements in $X$ (or elements of its vectors). For a vector $v \in \mathbb{Q}^d$ we define $\dim(v) \coloneqq d$, and for a matrix $W \in \mathbb{Q}^{d_1 \times d_2}$ we define $\dim(W) \coloneqq (d_1, d_2)$.

### Encoding and Bit-Length

Let $x \in \mathbb{Q}$ and let $\frac{p}{q} = x$, $p \in \mathbb{Z}, q \in \mathbb{N}$ be its *reduced form*, a *fractional representation* of $x$ is a bit-representation that encodes $p; q$ in separate - using any $O(\log n)$ integer encoding, and we assume all computations to use such representation. All fractions in this paper are in reduced form. For $q \in \mathbb{Q}$ we denote its fractional-representation bit-length by $\lambda_{\mathsf{f}}(q)$. For a vector $v \in \mathbb{Q}^d$ we define its bit-length $\lambda_{\mathsf{f}}(v) \coloneqq \Sigma_{i \in [d]} \lambda_{\mathsf{f}}(v(i))$. For a sequence or multiset of vectors $S = (v_i)_{i \in [n]}, M = \{\!\{w_i\}\!\}_{i \in [m]}$ we define their bit-length $\lambda_{\mathsf{f}}(S) \coloneqq \Sigma_{i \in [n]} \lambda_{\mathsf{f}}(v_i), \ \lambda_{\mathsf{f}}(M) \coloneqq \Sigma_{i \in [m]} \lambda_{\mathsf{f}}(w_i)$. For $d, k \in \mathbb{N}$ we define $\mathbb{Q}_k^d \coloneqq \{q : q \in \mathbb{Q}^d, \lambda_{\mathsf{f}}(q) \le k\}$ the dimension-$d$ rational vectors of bit-length no greater than $k$.

### Featured Graph

A (vertex) *featured graph* $G = \langle V(G), E(G), S, Z(G) \rangle$ is a 4-tuple being the usual undirected graph definition, with the addition of a *feature map* $Z(G) : V(G) \to S$ which maps each vertex to a value in some set $S$. For $v \in V(G)$ we define $N_G(v) \coloneqq \{w \in V(G) : vw \in E(G)\}$ the neighborhood of $v$, and we denote $Z(G)(v)$ also by $Z(G, v)$. We define the *order*, or *size*, of a graph $G$ to be the number of its vertices i.e. $|G| \coloneqq |V(G)|$. We denote the domain of graphs featured over a set $S$ by $\mathcal{G}_S$ and the set of all featured graphs by $\mathcal{G}_*$. In this paper we consider the domain of graphs with boolean input-features and denote it by $\mathcal{G}_{\mathcal{B}}$, that is, $\mathcal{G}_{\mathcal{B}} \coloneqq \{G \mid \forall v \in V(G) \ Z(G)(v) \in \{0, 1\}\}$. For a graph domain $\mathcal{G} \subseteq \mathcal{G}_*$, and $n \in \mathbb{N}$, we define $\mathcal{G}(n) \coloneqq \{G \in \mathcal{G} : |G| = n\}$ the graphs in $\mathcal{G}$ of size $n$. We denote the set of all feature maps that map to some set $T$ by $\mathcal{Z}_T$, and we denote the set of all feature maps by $\mathcal{Z}_*$. Let $\mathcal{G} \subseteq \mathcal{G}_*$, a mapping $f : \mathcal{G} \to \mathcal{Z}_*$ to new feature maps is called a *feature transformation*, and for $d \in \mathbb{N}$ a mapping $f : \mathcal{G} \to \mathbb{Q}^d$ is called a *graph embedding*.

### Multilayer Perceptron

A ReLU-activated Multilayer Perceptron (MLP) $F = (l_1, \ldots, l_m)$, $l_i = (w_i, b_i)$, of I/O dimensions $d_{in}; d_{out}$, and depth $m$, is a sequence of rational matrices $w_i$ and bias vectors $b_i$ such that

$$\dim(w_1)(2) = d_{in}, \dim(w_m)(1) = d_{out},$$

$$\forall i > 1 \ \dim(w_i)(2) = \dim(w_{i-1})(1), \ \forall i \in [m] \dim(b_i) = \dim(w_i)(1)$$

It defines a function $f_F(x)$, which we denote also by $F(x)$, such that

$$f_F(x) := w_m(...\text{ReLU}(w_2(\text{ReLU}(w_1(x) + b_1)) + b_2)...) + b_m, \ \ \text{ReLU}(x) := \max(0, x)$$

**Message-Passing Graph Neural Network**

A *Message Passing Graph Neural Network* (MP-GNN) of depth $m$ and dimensions $r_0, \{p_i, q_i, r_i\}_{i \in [m]}$

$$N = (\mathsf{mlp}_1, \text{agg}_1, \text{msg}_1), \ldots, (\mathsf{mlp}_m, \text{agg}_m, \text{msg}_m)$$

is a sequence of $m$ triplets, referred to as *layers*, such that for $i \in [m]$ layer $i$ comprises a message and aggregation functions and an MLP,

$$\text{msg}_i : \mathbb{Q}^{r_{i-1}} \times \mathbb{Q}^{r_{i-1}} \to \mathbb{Q}^{p_i}, \ \ \text{agg}_i : \left( \binom{\mathbb{Q}^{p_i}}{*} \right) \to \mathbb{Q}^{q_i}, \ \ \mathsf{mlp}_i : \mathbb{Q}^{r_{i-1}} \times \mathbb{Q}^{q_i} \to \mathbb{Q}^{r_i}$$

The message function is usually either $(x, y) \mapsto y$ or an MLP, but not necessarily. In this paper we will assume it is an MLP i.e. the more expressive among the two. The aggregation function is typically per-dimension sum; mean; or max, but can also be other functions that operate on a multiset and have a fixed output-dimension. The sequence of layers defines a feature transformation $f_N : \mathcal{G}_{\mathbb{Q}^{r_0}} \to \mathcal{Z}_{\mathbb{Q}^{r_m}}$ as follows: Let $G \in \mathcal{G}_{\mathbb{Q}^{r_0}}$ and $v \in V(G)$, then we define:

[1.] $v_N^{(0)} := N^{(0)}(G, v) := Z(G)(v)$ the initial value of $v$.

[2.] $\forall t \in [m] \quad v_N^{(t)} := N^{(t)}(G, v) := \mathsf{mlp}_t \left( v_N^{(t-1)}, \text{agg}_t(\{\{ \text{msg}_t(v_N^{(t-1)}, w_N^{(t-1)}) \mid w \in N_G(v)\}\}) \right)$
the value of $v$ after applying the first $t$ layers of $N$.

[3.] $N(G, v) := N^{(m)}(G, v)$ the final value of $v$.

When $G$ is clear from the context, we may use $v_N^{(t)}$ for $N^{(t)}(G, v)$. If in addition to its layers $N$ includes a readout step $R = (\mathsf{mlp}_R, \text{agg}_R)$, then it defines a graph embedding:

$$N(G) := \mathsf{mlp}_R(\text{agg}_R(\{\{v_N^{(m)} \mid v \in V(G)\}\}))$$

**Color Refinement**

Let $G \in \mathcal{G}_*$. For $t \geq 0$ and $v \in V(G)$ we define the *color of $v$ after $t$ iterations*, notated $\mathsf{cr}_G^{(t)}(v)$, inductively: The initial value of $v$ is its initial feature, that is, $\mathsf{cr}_G^{(0)}(v) := Z(G)(v)$, and for all $t > 0$ we define

$$\mathsf{cr}_G^{(t)}(v) := (\mathsf{cr}_G^{(t-1)}(v), \{\{\mathsf{cr}_G^{(t-1)}(w) \mid w \in N_G(v)\}\})$$

Maximum color-classes granularity is reached after at most $|G|$ iterations, hence we define the *color of $v$* to be $\mathsf{cr}_G(v) := \mathsf{cr}_G^{|G|}(v)$. We define the color of $G$ at iteration $t$, and overall, to be

$$\mathsf{cr}^{(t)}(G) := \{\{\mathsf{cr}^{(t)}(v) \mid v \in V(G)\}\}, \ \ \mathsf{cr}(G) := \mathsf{cr}^{|G|}(G)$$

## 3  Limited by Aggregation

We start with defining the aggregation class that is our main focus. Our characterization relates to the information complexity of an aggregation's output. We would like our definition to be general and capture a wide range of practical aggregations, and at the same time imply a significant upper bound on distinguishing-power. As we consider rational numbers, our characterization must account also for the common denominator of the values to-be-aggregated.

**Definition 3.1 (Logarithmic Aggregation).** *Let $d, d' \in \mathbb{N}$, $agg : \left( \binom{\mathbb{Q}^d}{*} \right) \to \mathbb{Q}^{d'}$ be an algorithm from a multiset of rational vectors to a single rational vector. We denote by $S_{agg} : \mathbb{N}^3 \to \mathbb{N}$ the output complexity of agg, depending on the number of vectors $n$, maximum bit-length $k$ of any vector, and the bit-length of the common denominator of all values.*

$$S_{agg}(n, k, \ell) := \max \left( \lambda_{\mathsf{f}}(agg(M)) : M \in \left( \binom{\mathbb{Q}_k^d}{n} \right), \lambda_{\mathsf{f}}(\mathrm{cd}(M)) \leq \ell \right)$$

*In addition, we denote by $S_{agg}^{\mathrm{cd}} : \mathbb{N}^3 \to \mathbb{N}$ the complexity of the common denominator of $n$ aggregations on subsets of a multiset of $n$ vectors, bit-length $k$ per vector, and multiset-common-denominator of bit-length $\ell$, that is,*

$$S_{agg}^{\mathrm{cd}}(n, k, \ell) := \max \left( \lambda_{\mathsf{f}}(\mathrm{cd}(M)) : M = \{\{agg(M_1), \ldots, agg(M_n)\}\}, \right.$$

$$M_i \subseteq M', M' \in \left( \binom{\mathbb{Q}_k^d}{n} \right), \lambda_{\mathsf{f}}(\mathrm{cd}(M')) \leq \ell \right)$$

*We say that agg is* logarithmic, *notated $\gamma(agg)$, if and only if for every $f, g : \mathbb{N} \to \mathbb{N}$ such that $f(n) = O(\log n), g(n) = O(\log n)$ it holds that:*

1. $S_{agg}(n, f(n), g(n)) = O(\log n)$.

2. $S_{agg}^{\mathrm{cd}}(n, f(n), g(n)) = O(\log n)$.

An example where $S_{\mathrm{agg}}$ is potentially non-logarithmic is the aggregation in Graph Attention Networks (Veličković et al., 2017) and Graph Transformers (Dwivedi & Bresson, 2020), which uses the *softmax* function - involving exponentiation by the input as well as division by graph-size dependent number. For the arithmetic mean, the condition on $S_{\mathrm{mean}}^{\mathrm{cd}}$ does not hold[2]: For $n \in \mathbb{N}$ define $M_n = \{1, 0, \ldots, 0\}, |m| = n$, and subsets $M'_{n,i} = \{1, 0, \ldots, 0\}, |M'_{n,i}| = i + 1, i \in [n-1]$, then we have $\mathrm{mean}(M'_{n,i}) = \frac{1}{i}$, hence by the *prime number theorem* (see for example (Hardy, 1999)) we have $\lim_{n \to \infty} \ln(\mathrm{cd}(\{\{M'_{n,i}\}\}_{i \in [n-1]})) = n$.

However, a vast range of aggregations is logarithmic. The following lemma provides useful general formulae for aggregations, which are logarithmic, and the subsequent example puts it to use in showing several commonly-used aggregations to be logarithmic. (See appendix for proofs details)

**Lemma 3.2.** *The following per-dimension aggregations $agg : \left( \binom{\mathbb{Q}^d}{*} \right) \to \mathbb{Q}^{d'}$ are logarithmic:*

1. $agg(M) := p_2(\Sigma_{x \in T} p_1(x)), T \subseteq M$, *for rational polynomials $p_1, p_2 : \mathbb{Q} \to \mathbb{Q}$.*

2. $agg(M) := (agg_1, \ldots, agg_a), \gamma(agg_i), a \in \mathbb{N}$. *That is, the concatenation of a fixed number of logarithmic aggregations.*

**Example 3.3.** *The following common aggregations are logarithmic:*

1. *sum.*

2. *Selection of $k$ elements, for a fixed $k \in \mathbb{N}$, by any criteria e.g. highest; lowest; quintile.*

3. *$k$-bins agg bin-aggregation, for a fixed $k \in \mathbb{N}$ and logarithmic aggregation agg.*

We proceed to quantify the distinguishing-power of MP-GNNs and $\mathrm{CR}^{(2)}$.

---

[2]Still, an exponential distinguishing-power upper bound can be shown. See Remark A.1 in the appendix.

**Distinguishing-Power.**

Let $\mathcal{G}$ be a graph domain, and $N$ be an MP-GNN, we define the *distinguishing-power* of $N$ on $\mathcal{G}$, $N_{\mathrm{dp}}(\mathcal{G})$, to be the number of vertices equivalence-classes that $N$ induces on $\mathcal{G}$. That is,

$$N_{\mathrm{dp}}(\mathcal{G}) := |\{N(G, v) : G \in \mathcal{G}, v \in V(G)\}|$$

Similarly, for $\mathrm{CR}^{(2)}$ we define

$$\mathrm{CR}^{(2)}_{\mathrm{dp}}(\mathcal{G}) := |\{\mathsf{cr}^{(2)}_G(v) : G \in \mathcal{G}, v \in V(G)\}|$$

For distinguishing between graphs, we define

$$\mathrm{N}_{\mathrm{gdp}}(\mathcal{G}) := |\{N(G) : G \in \mathcal{G}\}|, \ \mathrm{CR}^{(2)}_{\mathrm{gdp}}(\mathcal{G}) := |\{\mathsf{cr}^{(2)}(G) : G \in \mathcal{G}\}|$$

When the domain is defined with a size parameter, i.e. $\mathcal{G} = \mathcal{G}'(n)$ for some domain $\mathcal{G}'$, we may refer to the distinguishing-power as a function $f : \mathbb{N} \to \mathbb{N}$. For example, for $N_{\mathrm{dp}}$, $f(n) := N_{\mathrm{dp}}(\mathcal{G}'(n))$.

**Main Result**

Our fundamental result (Theorem 3.8) is that for any MP-GNN $N$ comprising (only) logarithmic aggregations, the distinguishing-power of $N$ is polynomial i.e.

$$N_{\mathrm{dp}}(\mathcal{G}_{\mathcal{B}}(n)) = \mathrm{poly}(n), \ N_{\mathrm{gdp}}(\mathcal{G}_{\mathcal{B}}(n)) = \mathrm{poly}(n)$$

To prove our main result, we take the following steps:

1. For any MP-GNN $N$, we define an information-complexity measure as a function of the graph size, $L_N : \mathbb{N} \to \mathbb{N}$, and observe that $N_{\mathrm{dp}}(\mathcal{G}_{\mathcal{B}}(n)), N_{\mathrm{gdp}}(\mathcal{G}_{\mathcal{B}}(n)) \leq 2^{L_N(n)}$. (Lemma 3.4)

2. We prove that for logarithmic-aggregations it holds that $L_N(n) = O(\log n)$. (Lemma 3.6)

3. We conclude the result from combining (1) and (2).

**Information Complexity.**

We measure the information conveyed by the aggregations in an MP-GNN's computation, as follows. Let $N = (\mathsf{mlp}_1, \mathrm{agg}_1, \mathsf{msg}_1), \ldots, (\mathsf{mlp}_m, \mathrm{agg}_m, \mathsf{msg}_m), (\mathsf{mlp}_R, \mathrm{agg}_R)$ be an MP-GNN, possibly with a final-readout layer. For a graph $G \in \mathcal{G}_{\mathcal{B}}$ and a vertex $v \in V(G)$ we define $I_N(G, v)$ to be the sequence of values produced by the aggregations operating on $(G, v)$, that is,

$$I_N(G, v) := \left(\mathrm{agg}_i \{\!\{ \mathsf{msg}_i(v^{(i-1)}, w^{(i-1)}) : w \in N_G(v) \}\!\}\right)_{i \in [m]}$$

, and we define

$$I_N(G) := \left(\mathrm{agg}_R \{\!\{ N(G, v) : v \in V(G) \}\!\}\right)$$

to be the information produced by the readout aggregation - if such exists. We define the complexity of $I_N$, $L_N : \mathbb{N} \to \mathbb{N}$, to be the maximum bit-length of $I_N$, that is, for a feature transformation

$$L_N(n) := \max\left(\lambda_{\mathsf{f}}(I_N(G, v)) : G \in \mathcal{G}_{\mathcal{B}}, \ |G| = n, v \in V(G)\right)$$

, and for a graph embedding

$$L_N^{(E)}(n) := \max\left(\lambda_{\mathsf{f}}(I_N(G)) : G \in \mathcal{G}_{\mathcal{B}}, \ |G| = n\right)$$

The reason for the specific definition of $L_N$ is the following key observation.

**Lemma 3.4.** *Let $N = (\mathsf{mlp}_1, agg_1, \mathrm{msg}_1), \ldots, (\mathsf{mlp}_m, agg_m, \mathrm{msg}_m), (\mathsf{mlp}_R, agg_R)$ be an MP-GNN, then*

$$N_{\mathrm{dp}}(\mathcal{G}_\mathcal{B}(n)) \leq 2^{L_N(n)+1}, \ N_{\mathrm{gdp}}(\mathcal{G}_\mathcal{B}(n)) \leq 2^{L_N^{(E)}(n)}$$

*Proof.* For $I_N(G, v)$, we show by induction on $m$ that

$$\forall G, G' \in \mathcal{G}_\mathcal{B} \ \forall v \in V(G) \ \forall v' \in V(G') \ Z(G)(v) = Z(G)(v') \wedge I_N(G, v) = I_N(G', v') \Rightarrow$$

$$N(G, v) = N(G', v')$$

For $m = 1$, $I_N(G, v) = agg_1\{\{\, \mathrm{msg}_1(v^{(0)}, w^{(0)}) \ : \ w \in N_G(v)\}\}$, hence $Z(G)(v) = Z(G')(v') \wedge I_N(G, v) = I_N(G', v') \Rightarrow \mathsf{mlp}_1(v^{(0)}, agg_1(\{\{\, \mathrm{msg}_1(v^{(0)}, w^{(0)}) \ : \ w \in N_G(v)\}\})) = \mathsf{mlp}_1(v'^{(0)}, agg_1(\{\{\, \mathrm{msg}_1(v'^{(0)}, w^{(0)}) \ : \ w \in N_{G'}(v')\}\})) \Rightarrow N(G, v) = N(G', v')$. Assuming correctness for $m = k$, we prove for $m = k + 1$. By the induction assumption, $Z(G)(v) = Z(G')(v') \wedge I_N(G, v) = I_N(G', v') \Rightarrow v^{(k)} = v'^{(k)}$. Also, $I_N(G, v) = I_N(G', v') \Rightarrow agg_{k+1}\{\{\, \mathrm{msg}_{k+1}(v^{(k)}, w^{(k)}) \ : \ w \in N_G(v)\}\} = agg_{k+1}\{\{\, \mathrm{msg}_{k+1}(v'^{(k)}, w^{(k)}) \ : \ w \in N_{G'}(v')\}\}$. Hence, $\mathsf{mlp}_{k+1}(v^{(k)}, agg_{k+1}(\{\{\, \mathrm{msg}_{k+1}(v^{(k)}, w^{(k)}) \ : \ w \in N_G(v)\}\})) = \mathsf{mlp}_{k+1}(v'^{(k)}, agg_{k+1}(\{\{\, \mathrm{msg}_{k+1}(v'^{(k)}, w^{(k)}) : w \in N_{G'}(v)\}\})) \Rightarrow N(G, v) = N(G', v')$.

For $I_N(G)$, $I_N(G) = I_N(G') \Rightarrow agg_R\{\{N(G, v) : v \in V(G)\}\} = agg_R\{\{N(G', v') : v' \in V(G')\}\} \Rightarrow \mathsf{mlp}_R(agg_R\{\{N(G, v) : v \in V(G)\}\}) = \mathsf{mlp}_R(agg_R\{\{N(G', v) : v \in V(G')\}\}) \Rightarrow N(G) = N(G')$.

Hence, the distinguishing-power of $N$ is upper-bounded by the number of possible values of $I_N$ (and initial feature, in case of a feature transformation), which in turn is upper-bounded exponentially by the maximum bit-length of $I_N$ (plus the 1 bit of initial feature, in case of a feature transformation). $\square$

The output of the aggregation of each layer in an MP-GNN depends on the output of the computation steps preceding it, hence, in order to calculate the total aggregations' output-complexity we need to calculate the intermediate-value complexity through the MP-GNN's computation steps. Before considering the complete MP-GNN's computation, we first observe the output-complexity of a single MLP. One may see why the following is true, as an MLP's effect on the magnitude of the input is limited - it is *Lipschitz continuous*, and also its effect (using ReLU activation) on the denominator of input values is bounded. Nevertheles, proof details can be found in the appendix.

**Lemma 3.5.** *Let $F$ be an MLP of input dimension $d$, we define $S_F : \mathbb{N} \to \mathbb{N}$ be the output-size complexity of $F$, that is,*

$$S_F(k) := \max(\lambda_{\mathsf{f}}(F(x)) : x \in \mathbb{Q}^d, \lambda_{\mathsf{f}}(x) = k)$$

*In addition, we denote by $S_{agg}^{\mathrm{cd}} : \mathbb{N}^3 \to \mathbb{N}$ the complexity of the common denominator of $n$ applications of $F$ on elements of a multiset of $n$ vectors, bit-length $k$ per vector, and multiset-common-denominator of bit-length $\ell$, that is,*

$$S_F^{\mathrm{cd}}(n, k, \ell) := \max \left(\lambda_{\mathsf{f}}(\mathrm{cd}(\{\{F(x_i)\}\}_{i \in [n]})) : \{\{x_i\}\}_{i \in [n]} = M, M \in \left(\!\!\binom{\mathbb{Q}_k^d}{n}\!\!\right), \lambda_{\mathsf{f}}(\mathrm{cd}(M)) \leq \ell\right)$$

*Then:*

1. $S_F(k) = O(k)$.

2. *For every $f : \mathbb{N} \to \mathbb{N}$ such that $f(n) = O(\log n)$ it holds that $S_F^{\mathrm{cd}}(n, k, f(n)) = O(\log n)$.*

We proceed to state our main lemma. Note that, referring to $L_N$, it considers the domain $\mathcal{G}_\mathcal{B}$ where the features are boolean thus their bit-length trivially does not exceed $O(\log n)$.

**Lemma 3.6.** *Let $N = (\mathsf{mlp}_1, agg_1, \mathrm{msg}_1), \ldots, (\mathsf{mlp}_m, agg_m, \mathrm{msg}_m), (\mathsf{mlp}_R, agg_R)$ be an MP-GNN, possibly with a final-readout layer, then: If all the aggregations are logarithmic then the total-information complexity of $N$ is logarithmic, formally*

$$\forall i \, \gamma(agg_i) \Rightarrow L_N(n) = O(\log n), \quad \forall i \, \gamma(agg_i) \wedge \gamma(agg_R) \Rightarrow L_N^{(E)}(n) = O(\log n)$$

*Proof.* For $l \in [m]$ we define $S_{N^{(l)}}^{\mathrm{msg}}, S_{N^{(l)}}^{\mathrm{agg}}, S_{N^{(l)}} : \mathbb{N} \to \mathbb{N}$ the complexities of intermediate outputs throughout the operation of $N$: The output of $\mathrm{msg}_l$, the output of $agg_l$, and the output of layer $l$ i.e. the output of $\mathsf{mlp}_l$. Note that these are not the complexities of the standalone functions - which we have defined and discussed earlier. Formally,

$$S_{N^{(l)}}^{\mathrm{msg}}(n) := \max \left( \lambda_{\mathsf{f}} \left( \mathrm{msg}_l(N^{(l-1)}(G, v), N^{(l-1)}(G, w)) \right) : G \in \mathcal{G}_\mathcal{B}, \ |G| = n, \ v \in V(G), \ w \in N_G(v) \right)$$

$$S_{N^{(l)}}^{\mathrm{agg}}(n) := \max \left( \lambda_{\mathsf{f}}(agg_l \{\{ \mathrm{msg}_l(N^{(l-1)}(G, v), N^{(l-1)}(G, w)) \mid w \in N_G(v) \}\}) : \right.$$

$$G \in \mathcal{G}_\mathcal{B}, \ |G| = n, \ v \in V(G) \Big), \quad S_{N^{(l)}}(n) := \max \left( \lambda_{\mathsf{f}}(N^{(l)}(G, v)) : G \in \mathcal{G}_\mathcal{B}, \ |G| = n, \ v \in V(G) \right)$$

In addition, we define $S_{N^{(l)}}^{\mathrm{cd,msg}}, S_{N^{(l)}}^{\mathrm{cd,agg}}, S_{N^{(l)}}^{\mathrm{cd}} : \mathbb{N} \to \mathbb{N}$ the complexities of the common denominator of intermediate values across the vertices, throughout the operation of $N$. Formally,

$$S_{N^{(l)}}^{\mathrm{cd,msg}}(n) := \max \left( \lambda_{\mathsf{f}} \left( \mathrm{cd} \left( \{\{ \mathrm{msg}_l(N^{(l-1)}(G, v), N^{(l-1)}(G, w)) \mid v \in V(G), \ w \in N_G(v) \}\} \right) \right) : \right.$$

$$G \in \mathcal{G}_\mathcal{B}, \ |G| = n \Big), \quad S_{N^{(l)}}^{\mathrm{cd,agg}}(n) := \max \left( \lambda_{\mathsf{f}} \left( \mathrm{cd} \left( \{\{ agg_l \{\{ \mathrm{msg}_l(N^{(l-1)}(G, v), N^{(l-1)}(G, w)) \mid \right. \right. \right.$$

$$w \in N_G(v) \}\} : v \in V(G) \}\} \Big) \Big) : G \in \mathcal{G}_\mathcal{B}, \ |G| = n \Big)$$

$$S_{N^{(l)}}^{\mathrm{cd}}(n) := \max \left( \lambda_{\mathsf{f}} \left( \mathrm{cd}(\{\{ N^{(l)}(G, v) : v \in V(G) \}\}) \right) : G \in \mathcal{G}_\mathcal{B}, \ |G| = n \right)$$

We prove by induction on $l$ that $\forall \, l \in [m] \ S_{N^{(l)}}^{\mathrm{msg}}, S_{N^{(l)}}^{\mathrm{agg}}, S_{N^{(l)}} = O(\log n)$. As the complexity of a sum of a fixed number of $O(\log n)$-complexity functions is $O(\log n)$, and by definition

$$L_N(n) = \max \left( \Sigma_{l \in [m]} \lambda_{\mathsf{f}}(agg_l \{\{ \mathrm{msg}_l \left( N^{(l-1)}(G, v), N^{(l-1)}(G, w) \right) \}\}_{w \in N_G(v)}) : \right.$$

$$G \in \mathcal{G}_\mathcal{B}, \ |G| = n, \ v \in V(G) \Big)$$

, by proving the induction we would have proven that $L_N(n) = \log n$. As part of the induction proof we also proof by induction on $l$ that $\forall \, l \in [m] \ S_{N^{(l)}}^{\mathrm{cd,msg}}, S_{N^{(l)}}^{\mathrm{cd,agg}}, S_{N^{(l)}}^{\mathrm{cd}} = O(\log n)$. For $l = 1$, by the initial features all being in $\{0, 1\}$, clearly $\exists c \in \mathbb{N} : S_{N^{(1)}}^{\mathrm{msg}}(n), S_{N^{(1)}}^{\mathrm{cd,msg}}(n) \leq c$ , hence trivially $S_{N^{(1)}}^{\mathrm{msg}}(n), S_{N^{(1)}}^{\mathrm{cd,msg}}(n) = O(\log n)$. Then, by assumption on $agg_1$ we have $S_{N^{(1)}}^{\mathrm{agg}}(n), S_{N^{(1)}}^{\mathrm{cd,agg}}(n) = O(\log n)$. By Lemma 3.5 and since the complexity of a composition of a function of complexity $O(n)$ over a function of complexity $O(\log n)$ is $O(\log n)$, we have that $S_{N^{(1)}}(n), S_{N^{(1)}}^{\mathrm{cd}}(n) = O(\log n)$. Assuming correctness for $l = k < m$ we prove for $l = k + 1$. By by the induction assumption on $S_{N^{(k)}}$ and by Lemma 3.5, $S_{N^{(k+1)}}^{\mathrm{msg}}(n)$ is the complexity of a composition of an $O(n)$-complexity function over the concatenation of two $O(\log n)$-complexity functions, which is $O(\log n)$. Also,

by the induction assumption on $S_{N^{(k)}}^{\text{cd}}$ and by Lemma 3.5 $S_{N^{(k+1)}}^{\text{cd,msg}}(n) = O(\log n)$. By assumption on $agg_{k+1}$, and by $S_{N^{(k+1)}}^{\text{msg}}(n), S_{N^{(k+1)}}^{\text{cd,msg}}(n) = O(\log n)$, we have $S_{N^{(k+1)}}^{\text{agg}}(n), S_{N^{(k+1)}}^{\text{cd,agg}}(n) = O(\log n)$. Finally, by the latter and by Lemma 3.5 we have $S_{N^{(k+1)}}(n), S_{N^{(k+1)}}^{\text{cd}}(n) = O(\log n)$.

For $L_N^{(E)}$, by definition $L_N^{(E)}(n) \coloneqq \max\left(\lambda_{\mathfrak{f}}(agg_R\{\{N(G, v)\}\}_{v \in V(G)}) : G \in \mathcal{G}_{\mathcal{B}}, \ |G| = n\right)$, hence by $S_{N^{(m)}}(n), S_{N^{(m)}}^{\text{cd}}(n) = O(\log n)$, and by assumption on $agg_R$, we have $L_N^{(E)}(n) = O(\log n)$ □

**Remark 3.7.** *The line of proof of Lemma 3.6 works for every message and combination functions with output-size complexity, and outputs-common-denominator complexity, $O(n)$ (with $n$ being the function's input size, as well as the outputs-multiset size), not only for ReLU-activated MLPs. Hence, the guarantee that $L_N(n) = O(\log n)$ (with $n$ being the input-graph size), and subsequently Theorem 3.8, hold for all MP-GNNs architectures comprising message and combination functions that have these properties.*

**Theorem 3.8.** *Let $N = (\mathsf{mlp}_1, agg_1, \text{msg}_1), \ldots, (\mathsf{mlp}_m, agg_m, \text{msg}_m), (\mathsf{mlp}_R, agg_R)$ be an MP-GNN, possibly with a final-readout layer, then: If all aggregations are logarithmic then the distinguishing-power of $N$ is polynomial. Formally,*

$$\forall i \ \gamma(agg_i) \Rightarrow N_{\text{dp}}(\mathcal{G}_{\mathcal{B}}(n)) = \text{poly}(n), \ \forall i \ \gamma(agg_i) \wedge \gamma(agg_R) \Rightarrow N_{\text{gdp}}(\mathcal{G}_{\mathcal{B}}(n)) = \text{poly}(n)$$

*Proof.* By assumption and Lemma 3.6 we have $L_N(n) = O(\log n)$, $L_N^{(E)}(n) = O(\log n)$, hence by Lemma 3.4 we have $N_{\text{dp}}(\mathcal{G}_{\mathcal{B}}(n)) = 2^{O(\log n)} = \text{poly}(n)$, $N_{\text{gdp}}(\mathcal{G}_{\mathcal{B}}(n)) = 2^{O(\log n)} = \text{poly}(n)$. □

**Comparison to Color Refinement**

The absolute-terms upper bounds in Theorem 3.8 are meaningful on their own, considering that the number of non-isomorphic graphs is super-exponential, $|\mathcal{G}_{\mathcal{B}}(n)| = 2^{\Omega(n^2 - n \log n)}$. In previous studies, the distinguishing-power of MP-GNNs has been compared to the distinguishing-power of Color Refinement (CR), where it was shown to either match it or not, depending on the setting, with no quantification given for the gap in the latter case. As CR is meaningful and well-studied, we proceed to put Theorem 3.8 in its perspective. We compare the distinguishing-power of logarithmic-aggregations MP-GNNs to the distinguishing-power of merely two iterations of CR, i.e. to $\text{CR}_{\text{dp}}^{(2)}(\mathcal{G}_{\mathcal{B}})$. We observe the following.

**Lemma 3.9.** *Let $n \in \mathbb{N}$, then $CR_{\text{dp}}^{(2)}(\mathcal{G}_{\mathcal{B}}(2n + 3)) \geq \binom{2n}{n}$, $CR_{\text{gdp}}^{(2)}(\mathcal{G}_{\mathcal{B}}(2n + 3)) \geq \binom{2n}{n}$*

*Proof.* We look at a two-level star graph of size $2n+3$, where we denote the center by $v$, the vertices of the first level by $u_i, i \in [n]$ and those of the second level by $w_i, i \in [n]$. We define

$$\mathcal{K}_n \coloneqq \{(k_0, \ldots k_n), k_i \in [0..n], \text{sum}(k_i) = n\}$$

all the possible choices, with repetition, of $n$ elements from $n + 1$ types. For $K \in \mathcal{K}_n$ we define $G_K \in \mathcal{G}_{\mathcal{B}}$ to be the graph where $v$ is connected to all $u$'s, and there are $k_i$ of the $u$'s that are connected to $i$ of the $w$'s. In other words $v$ has $k_i$ neighbors of degree $i$ (+1). In addition, $v$ is connected to two vertices $s_1, s_2$ to make its degree higher than any of the $u$'s and $w$'s. Formally, $G_K$ is defined as follows: $V(G) = \{v\}, \{s_1, s_2\}\{u_1, \ldots, u_n\}, \{w_1, \ldots, w_n\}$,

$$E(G) = \{vs_1, vs_2\}, \{vu_i : i \in [n]\}, \{u_i w_j : j \in [n], i > \Sigma_{h=0}^{j-1} k_h\}, \ \forall x \in V(G) \ Z(G)(x) = 1$$

For example, let $K \in \mathcal{K}_4, K = \{1, 2, 0, 1, 0\}$, then $G_K$'s vertices and edges are $V(G_K) = \{v, s_1, s_2\}, \{u_i\}_{i \in [4]}, \{w_i\}_{i \in [4]}, E(G_K) = \{vs_1, vs_2\}, \{vu_i : i \in [4]\}, \{u_2 w_1, u_3 w_1, u_4 w_1, u_4 w_2, u_4 w_3\}$. See Section A.1 in the appendix, for an illustration.

For $n \in \mathbb{N}$ we define $\mathcal{G}_{\mathcal{K}}(2n+3) \coloneqq \{G_K : K \in \mathcal{K}_n\} \subset \mathcal{G}_{\mathcal{B}}$ the set of all graphs of the form above, of size $2n+3$. Observe that $|\mathcal{G}_{\mathcal{K}}(2n+3)| = \binom{2n}{n}$, as it is the number of options to choose with repetition $n$ elements - the number of $u$ vertices - out of $n$ possible types - the possible number of neighbors. In addition, the difference in connectivity of the $u$ layer and the $w$ layer, between every $G_K \neq G_{K'} \in \mathcal{G}_{\mathcal{K}}(2n+3)$, implies that the color of $isunique(v) \neq \mathsf{cr}^{(2)}_{G'}(v)$. Formally, $\left( \forall n \in \mathbb{N} \ \forall G \neq G' \in \mathcal{G}_{\mathcal{K}}(2n+3) \ \ \exists i : |\{j : |N_G(w_j)| = i\}| \neq |\{j : |N_{G'}(w_j)| = i\}| \right) \Rightarrow \left( \forall n \in \right.$
$\left. \mathbb{N} \ \forall G \neq G' \in \mathcal{G}_{\mathcal{K}}(2n+3) \ \ \mathsf{cr}^{(2)}_G(v) \neq \mathsf{cr}^{(2)}_{G'}(v) \right)$, hence $\mathrm{CR}^{(2)}_{\mathrm{dp}}(\mathcal{G}_{\mathcal{B}}(2n+3)) \geq \binom{2n}{n}$. Finally, the color of $v$ is unique also compared to that of $u$ and $w$ in all graphs (of size $(2n+3)$), as their degree is $\leq n+1$. Hence, by definition of $\mathsf{cr}$ we have $\forall n \in \mathbb{N} \ \forall G, G' \in \mathcal{G}_{\mathcal{K}}(2n+3) \ \ \mathsf{cr}^{(2)}(G) \neq \mathsf{cr}^{(2)}(G')$, hence $\mathrm{CR}^{(2)}_{\mathrm{gdp}}(\mathcal{G}_{\mathcal{B}}(2n+3)) \geq \binom{2n}{n}$ $\qquad \square$

Combined with Lemma 3.4, we arrive at the following sufficient condition for an MP-GNN having weaker distinguishing-power than $\mathrm{CR}^{(2)}$, and combined with Theorem 3.8 we have the following measure of the gap between the power of logarithmic-aggregations MP-GNNs and that of $\mathrm{CR}^{(2)}$.

**Corollary 3.10.** *Let* $N = (\mathsf{mlp}_1, agg_1, \mathsf{msg}_1), \ldots, (\mathsf{mlp}_m, agg_m, \mathsf{msg}_m), (\mathsf{mlp}_R, agg_R)$ *be an MP-GNN, possibly with a final-readout layer, then:*

1. *If there exists* $n \in \mathbb{N}$ *such that* $L_N(2n+3) < \left\lceil \log \binom{2n-1}{n-1} \right\rceil$ *then there are vertices that are distinguishable by* $CR^{(2)}$ *and not by* $N$. *Formally,* $\exists n \in \mathbb{N} : L_N(2n+3) < \left\lceil \log \binom{2n-1}{n-1} \right\rceil \Rightarrow$

$$\exists G, G' \in \mathcal{G}_{\mathcal{B}}, \ v \in V(G), \ v' \in V(G') : \mathsf{cr}^{(2)}_G(v) \neq \mathsf{cr}^{(2)}_{G'}(v') \wedge N(G, v) = N(G', v')$$

*In particular,* $\forall i \ \gamma(agg_i) \Rightarrow \lim_{n \to \infty} \frac{N_{\mathrm{dp}}(\mathcal{G}_{\mathcal{B}}(n))}{CR^{(2)}_{\mathrm{dp}}(\mathcal{G}_{\mathcal{B}}(n))} = 0$

2. *For distinguishing graphs,*

$$\exists n \in \mathbb{N} : L^{(E)}_N(2n+3) < \left\lceil \log \binom{2n-1}{n-1} \right\rceil \Rightarrow \exists G, G' \in \mathcal{G}_{\mathcal{B}} : \mathsf{cr}^{(2)}(G) \neq \mathsf{cr}^{(2)}(G') \wedge N(G) = N(G')$$

$$\forall i \ \gamma(agg_i) \wedge \gamma(agg_R) \Rightarrow \lim_{n \to \infty} \frac{N_{\mathrm{gdp}}(\mathcal{G}_{\mathcal{B}}(n))}{CR^{(2)}_{\mathrm{gdp}}(\mathcal{G}_{\mathcal{B}}(n))} = 0$$

## 4 Concluding Remarks

We have introduced an output-size complexity property for aggregation functions, satisfied by most of the reasonable aggregations that do not involve exponentiation or division by a graph-size-dependent value, and proved that it has the effect of restricting MP-GNN models to distinguish merely a polynomial number of equivalence classes. This applies both to distinguishing between vertices and distinguishing between graphs. Given that the number of non-isomorphic graphs is super-exponential, $2^{\Omega(n^2 - n \log n)}$, that bound is significant.

We proceeded to take a familiar perspective and considered the well-studied distinguishing-power of the Color Refinement algorithm, already known to upper-bound all MP-GNNs, as a reference point. We have observed that $\mathrm{CR}^{(2)}$, i.e. merely 2 iterations of CR, is not only stronger than our general class of MP-GNNs, making $\mathrm{CR}^{(1)}$ a tight bound [3] for it, but is relatively infinitely stronger,

as it is at least exponential. This is in stark contrast to non-uniform distinguishing-power results (Xu et al., 2019; Morris et al., 2019; Aamand et al., 2022), as well as to uniform results for recurrent MP-GNNs (Rosenbluth & Grohe, 2025).

A consequence of our results is that every function, in every function-class that is subsumed by logarithmic-aggregations MP-GNNs, does not distinguish more than a polynomial number of equivalence-classes.

To practice, an immediate implication of our results is that if the target function assumes (on the graph domain) a greater-than polynomial number of values in the graph size then it is simply impossible for a logarithmic-aggregations MP-GNN model to even distinguish between all vertices or graphs that are assigned a different value by the function, let alone assign them the specific function's values.

While we focus on MP-GNNs with ReLU-activated MLPs for message and combination functions, our results hold for all MP-GNNs comprising message and combination functions with output and outputs-common-denominator size-complexities of $O(n)$.

Our goal is to understand fundamental expressivity bounds of MP-GNNs - regardless of aggregations specifics. To that end, the following remain open for further research:

1. We have not addressed aggregations that have enough output bits to represent the number of all possible graphs. There, (full) CR distinguishing-power is potentially given for free - the aggregation function can simply implement CR, and the question to study is that of expressivity i.e. approximating a target function - computing a specific value for each vertex or graph. It is clear that the fixed number of MLP-runs in an MP-GNN with rational weights is a limiting factor, as a runtime-complexity upper bound of

$$O(n^2 \cdot \text{aggregation output-size complexity}) + \text{total aggregation-runtime}$$

   on the run of any MP-GNN should be relatively straightforward. However, a tighter bound, or perhaps one in terms other than runtime-complexity, for the expressivity of MP-GNNs with arbitrary computable aggregations, can be interesting.

2. We have not analyzed the output-size complexity of softmax aggregation. Potentially it is linear, rather than logarithmic , as it involves exponentiation by the input, however, a careful examination of the computation - taking into account the normalization and the actual algorithm for computing it - may prove otherwise.

3. We have not analyzed the output-size complexity of MLPs with non-ReLU activations such as *sigmoid* or *tanh*, which without considering computability have been shown to increase the distinguishing-power of MP-GNNs (Khalife & Basu, 2023). There again, an exponentiation by the input is involved, potentially leading to an exponential rather than linear output-size complexity of the MLP and higher distinguishing-power, yet further analysis is required for a clear characterization.

---

[3]when the graph diameter does not exceed the number of CR iterations.

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

## A  Limited By Aggregation

**Remark A.1.** *Let $q_c^{(N)}$ be the common denominator of all the parameters of a mean-aggregation MP-GNN $N$, and for $n \in \mathbb{N}$ define $q_c^{(n)} := \mathrm{cd}(\{\frac{1}{i+1}\}_{i \in [n-1]})$, then it is not difficult to show that $q_c^{(N)} q_c^{(n)}$ is a common denominator of all the computations of $N$ on graphs of size $n$. Noting that $q_c^{(n)} = O(3^n)$ (Hanson, 1972), the line of proof of Lemma 3.6 would lead to $L_N(n) = O(n)$ and eventually to an exponential distinguishing-power upper bound. Such bound is meaningful as it is lower than the super-exponential number of non-isomorphic graphs.*

To prove Lemma 3.2, we first prove the following two lemmas.

**Lemma A.2.** *The application of a polynomial does not affect the asymptotic bit-length complexity. Formally, let $p : \mathbb{Q} \to \mathbb{Q}$ be a rational polynomial and define $S : \mathbb{N} \to \mathbb{N}$, $S(n) := \max(\lambda_{\mathfrak{f}}(p(x)) : \lambda_{\mathfrak{f}}(x) \leq (n))$, then $S(n) = O(n)$.*

*Proof.* Let $p(x) = \Sigma_{i=0}^k \frac{a_i}{b_i} x^i, a_i \in \mathbb{Z}, b_i \in \mathbb{N}$, and let $y \in \mathbb{Z}, z \in \mathbb{N}, n \in \mathbb{N}$ such that $\lambda_{\mathfrak{f}}(\frac{y}{z}) = n$. Then,

$$p(\frac{y}{z}) = \Sigma_{i=0}^k \frac{a_i y^i z^k \prod_{j=0}^k b_j}{z^i b_i z^k \prod_{j=0}^k b_j} = \frac{1}{z^k \prod_{j=0}^k b_j} \Sigma_{i=0}^k a_i y^i z^{k-i} \prod_{j \neq i} b_j$$

Hence, $\lambda_{\mathfrak{f}}(p(\frac{y}{z})) \leq \log(z^k \prod_{j=0}^k b_j) + \log(\Sigma_{i=0}^k |a_i||y|^i z^{k-i} \prod_{j \neq i} b_j) \leq \log(z^k \prod_{j=0}^k b_j) + \Sigma_{i=0}^k \log(|a_i||y|^i z^{k-i} \prod_{j \neq i} b_j) = O(\log(z^k)) + \Sigma_{i=0}^k O(\log(|y|^i z^{k-i})) \leq O(\log n) + k^2 O(\log n) = O(\log n)$  □

**Lemma A.3.** *The bit-length of the sum of $n$ elements of length $O(\log n)$ and common-denominator-length $O(\log n)$ is $O(\log n)$.*

*Proof.* Let $n \in \mathbb{N}$, $M \in \left( \binom{\mathbb{Q}}{n} \right)$, $M = \{\{\frac{p_i}{q_i}, p_i \in \mathbb{Z}, q_i \in \mathbb{N}\}\}_{i \in [n]}, \lambda_{\mathfrak{f}}(\mathrm{cd}(M) = O(\log n)$, and define $q := \mathrm{cd}(M)$, then

$$\lambda_{\mathfrak{f}}(\Sigma_{i \in [n]} \frac{p_i}{q_i}) = \lambda_{\mathfrak{f}}(\frac{1}{q} \Sigma_{i \in [n]} \frac{p_i q}{q_i}) \leq O(\log n) + O(\log(n 2^{O(\log n)} 2^{O(\log n)})) = O(\log n)$$

□

**Lemma 3.2.** *The following per-dimension aggregations $agg : \left( \binom{\mathbb{Q}^d}{*} \right) \to \mathbb{Q}^{d'}$ are logarithmic:*

1. *$agg(M) := p_2(\Sigma_{x \in T} p_1(x)), T \subseteq M$, for rational polynomials $p_1, p_2 : \mathbb{Q} \to \mathbb{Q}$.*

2. *$agg(M) := (agg_1, \ldots, agg_a), \gamma(agg_i), a \in \mathbb{N}$. That is, the concatenation of a fixed number of logarithmic aggregations.*

*Proof.* As these aggregations operate on vectors per-dimension, it is enough to show that they are logarithmic when operating on multisets of scalars.

1. Let $p_1, p_2 : \mathbb{Q} \to \mathbb{Q}$ be rational polynomials and let $f, g : \mathbb{N} \to \mathbb{N}$ such that $f(n), g(n) = O(\log n)$. Let $n \in \mathbb{N}, M \in \left( \binom{\mathbb{Q}}{n} \right)$, $M = \{\{x_1, \ldots, x_n\}\}, \lambda_{\mathfrak{f}}(x_i) \leq f(n), \lambda_{\mathfrak{f}}(\mathrm{cd}(M)) \leq g(n)$ be a multiset of elements of bit-length at most $f(n)$ per-element, and a common denominator of bit-length at most $g(n)$, and let $T \subseteq M, T = \{x'_1, \ldots, x'_m\}$. By Lemma A.2 we have

that $\forall i\ \lambda_{\mathsf{f}}(p_1(x_i')) = O(\log n)$. Hence, by Lemma A.3 we have that $\lambda_{\mathsf{f}}(\Sigma_{i\in[m]}p_1(x_i')) = O(\log n)$. Hence, by Lemma A.2 we have that $\lambda_{\mathsf{f}}(p_2(\Sigma_{i\in[m]}p_1(x_i'))) = O(\log n)$. Hence, $S_{\mathrm{agg}} = O(\log n)$.

For $S_{\mathrm{agg}}^{\mathrm{cd}}$, define $q_{\mathrm{cd}} := \mathrm{cd}(M)$, assume $p_1(x) = \Sigma_{i=0}^{k_1}\frac{a_i}{b_i}x^i, p_2(x) = \Sigma_{i=0}^{k_2}\frac{r_i}{s_i}x^i,\ b_i \in \mathbb{N}, s_i \in \mathbb{N}, a_i \in \mathbb{Z}, r_i \in \mathbb{Z}$, define $q_{p_1} := q_{\mathrm{cd}}^{k_1}\prod_{i=0}^{k_1}b_i$, and define $q := q_{p_1}^{k_2}\prod_{i=0}^{k_2}s_i$. Observe that:

1. $q_{p_1}$ is a common denominator for all applications of $p_1$ on values that are commonly denominated by $q_{\mathrm{cd}}$. That is, let $\frac{y}{z} \in \mathbb{Q}$ such that $\frac{q_{\mathrm{cd}}}{z} \in \mathbb{N}$, then there exists $y' \in \mathbb{Z}$ such that $p_1(\frac{y}{z}) = \frac{y'}{q_{p_1}}$. This is because $\forall i\ \frac{q_{p_1}}{z^i b_i} = (\frac{q_{\mathrm{cd}}}{z})^i q_{\mathrm{cd}}^{k_1-i}\prod_{j\neq i}b_j \in \mathbb{N}$.

2. Similarly, $q$ is a common denominator for all applications of $p_2$ on values that are commonly denominated by $q_{p_1}$. That is, let $\frac{y}{z} \in \mathbb{Q}$ such that $\frac{q_{p_1}}{z} \in \mathbb{N}$, then there exists $y' \in \mathbb{Z}$ such that $p_2(\frac{y}{z}) = \frac{y'}{q}$. This is because $\forall i\ \frac{q}{z^i s_i} = (\frac{q_{p_1}}{z})^i q_{p_1}^{k_2-i}\prod_{j\neq i}s_j \in \mathbb{N}$.

By (1) we have that $\forall W \subseteq M\ \ \frac{\Sigma_{x\in W}p_1(x)}{q_{p_1}} \in \mathbb{Z}$. Then, by (2) we have that $\forall W \subseteq M\ \ \frac{p_2(\Sigma_{x\in W}p_1(x))}{q} \in \mathbb{Z}$. Hence, we have that $\forall U \subseteq \left(\!\!\binom{M}{*}\!\!\right)\ \ \mathrm{cd}(\{\!\{p_2(\Sigma_{x\in W}p_1(x))\}\!\}_{W\in U}) \leq q$. Hence, $\lambda_{\mathsf{f}}(\mathrm{cd}(\{\!\{p_2(\Sigma_{x\in W}p_1(x))\}\!\}_{W\in U})) \leq \lambda_{\mathsf{f}}(q) = O(\log(q_{p_1}^{k_2}\prod_{i=0}^{k_2}s_i)) = O(\log(q_{p_1}^{k_2})) = O(\log(q_{\mathrm{cd}}^{k_1}\prod_{i=0}^{k_1}b_i)) = O(\log(q_{\mathrm{cd}}^{k_1})) = O(k_1\log(q_{\mathrm{cd}})) = O(\log n)$. Hence, $S_{\mathrm{agg}}^{\mathrm{cd}} = O(\log n)$.
[2.] $\lambda_{\mathsf{f}}(\mathrm{agg}) = \Sigma_{i\in[a]}\lambda_{\mathsf{f}}(\mathrm{agg}_i) = aO(\log n) = O(\log n)$. As for $S_{\mathrm{agg}}^{\mathrm{cd}}$, let $U \subseteq \left(\!\!\binom{M}{*}\!\!\right)$, define $\mathrm{cd}_i := \mathrm{cd}(\{\!\{\mathrm{agg}_i(T)\}\!\}_{T\in U})$, and define $\mathrm{cd}_{\mathrm{agg}} := \mathrm{cd}(\{\!\{\mathrm{agg}(T)\}\!\}_{T\in U})$. Then $\mathrm{cd}_{\mathrm{agg}} \leq \prod_{i\in[a]}\mathrm{cd}_i$, hence $\lambda_{\mathsf{f}}(\mathrm{cd}_{\mathrm{agg}}) \leq \lambda_{\mathsf{f}}(\prod_{i\in[a]}\mathrm{cd}_i) = O(\log(\prod_{i\in[a]}\mathrm{cd}_i)) = O(aO(\log n)) = O(\log n)$.

$\square$

**Example 3.3.** *The following common aggregations are logarithmic:*

1. *sum.*

2. *Selection of $k$ elements, for a fixed $k \in \mathbb{N}$, by any criteria e.g. highest; lowest; quintile.*

3. *$k$-bins agg bin-aggregation, for a fixed $k \in \mathbb{N}$ and logarithmic aggregation agg.*

*Proof.* 1. By Lemma 3.2(1), setting $T = M$, $p_1(x) = x$, $p_2(x) = x$, we have that sum is logarithmic.

2. Defining $\mathrm{agg}_i$ to be the selection of the $i^{\mathrm{th}}, i \in [k]$ element (by whichever criteria) we have that $\mathrm{agg}_i$ is logarithmic. Then, by Lemma 3.2(2) we have that $(\mathrm{agg}_1,\ldots,\mathrm{agg}_k)$ is logarithmic i.e. the selection of the $k$ elements is logarithmic.

3. Note that by agg being logarithmic when applied to $M$, it is logarithmic when applied to any $T \subseteq M$. Defining $T_i, i \in [k]$ to be the elements in bin $k$, and defining $\mathrm{agg}_i := \mathrm{agg}(T_i)$ we have that $\mathrm{agg}_i$ is logarithmic and by Lemma 3.2(2) $(\mathrm{agg}_1,\ldots,\mathrm{agg}_k)$ is logarithmic i.e. the sequence of aggregated bins is logarithmic.

$\square$

**Lemma 3.5.** *Let $F$ be an MLP of input dimension $d$, we define $S_F : \mathbb{N} \to \mathbb{N}$ be the output-size complexity of $F$, that is,*

$$S_F(k) := \max(\lambda_{\mathsf{f}}(F(x)) : x \in \mathbb{Q}^d, \lambda_{\mathsf{f}}(x) = k)$$

*In addition, we denote by $S_{agg}^{\mathrm{cd}} : \mathbb{N}^3 \to \mathbb{N}$ the complexity of the common denominator of $n$ applications of $F$ on elements of a multiset of $n$ vectors, bit-length $k$ per vector, and multiset-common-denominator of bit-length $\ell$, that is,*

$$S_F^{\mathrm{cd}}(n, k, \ell) := \max\left(\lambda_{\mathsf{f}}(\mathrm{cd}(\{\{F(x_i)\}\}_{i \in [n]})) : \{\{x_i\}\}_{i \in [n]} = M, M \in \left(\!\!\binom{\mathbb{Q}_k^d}{n}\!\!\right), \lambda_{\mathsf{f}}(\mathrm{cd}(M)) \le \ell\right)$$

*Then:*

1. $S_F(k) = O(k)$.

2. *For every* $f : \mathbb{N} \to \mathbb{N}$ *such that* $f(n) = O(\log n)$ *it holds that* $S_F^{\mathrm{cd}}(n, k, f(n)) = O(\log n)$.

*Proof.* 1. Assume $F = (l_1, \dots, l_m), l_i = (w_i, b_i)$, $\dim(w_i) = (d_{i+1}, d_i)$. Define $S_F(k)$ to be the output-size complexity of a single layer $l$, i.e. $S_F(k) := \max(\lambda_{\mathsf{f}}(\mathrm{ReLU}(w_l x + b_l)) : x \in \mathbb{Q}^{d_l}, \lambda_{\mathsf{f}}(x) = k)$, we show that $S_{F_l}(n) = O(n)$. As it is straightforward that the output-size complexity of a composition of a fixed number of functions of linear output-size complexity is linear, proving $\forall l \; S_{F_l}(n) = O(n)$ will prove $S_F(n) = O(n)$.

Assume w.l.o.g that $l = l_1$. For $i \in [d_2], j \in [d_1]$ let $\frac{p_{i,j}^{(w)}}{q_{i,j}^{(w)}} = w_1(i, j), p_{i,j}^{(w)} \in \mathbb{Z}, q_{i,j}^{(w)} \in \mathbb{N}$, and for $i \in [d_2]$ let $\frac{p_i^{(b)}}{q_i^{(b)}} = b_1(i), p_i^{(v)} \in \mathbb{Z}, q_i^{(b)} \in \mathbb{N}$. Define $q_c^{(w)} := \prod_{i \in [d_2], j \in [d_1]} q_{i,j}^{(w)} \prod_{i \in [d_2]} q_i^{(b)}$, a common denominator of all the parameters. Let $n \in \mathbb{N}, x \in \mathbb{Q}^{d_1}, \lambda_{\mathsf{f}}(x) = n$. For $i \in [d_1]$ let $\frac{p_i^{(x)}}{q_i^{(x)}} = x(i), p_i^{(x)} \in \mathbb{Z}, q_i^{(x)} \in \mathbb{N}$, and define $q_c^{(x)} := \prod_{i \in [d_1]} q_i^{(x)}$ a common denominator of the input values. For $i \in [d_2]$ let $\frac{p_i^{(r)}}{q_i^{(r)}} = \mathrm{ReLU}\left((w_1 x)(i) + b_1(i)\right), p_i^{(r)} \in \mathbb{Z}, q_i^{(r)} \in \mathbb{N}$, then

$$\lambda_{\mathsf{f}}(q_i^{(r)}) \le \lambda_{\mathsf{f}}(q_c^{(w)} \cdot q_c^{(x)}) = O(\log(q_c^{(w)}) + \log(q_c^{(x)})) = O(\log(q_c^{(x)})) = O(\Sigma_{i \in [d_1]} \log(q_i^{(x)}))$$
$$= O(\lambda_{\mathsf{f}}(x)) \quad \text{(A.1)}$$

Define $p_{\max}^{(w)} := \max(\left|p_{i,j}^{(w)}\right| : i \in [d_2], j \in [d_1])$, then

$$\lambda_{\mathsf{f}}(p_i^{(r)}) \le \lambda_{\mathsf{f}}(q_i^{(r)}(p_i^{(b)} + \Sigma_{j \in [d_1]} p_{i,j}^{(w)} p_j^{(x)})) = O\left(\log(q_i^{(r)}) + \log(\left|p_i^{(b)}\right| + p_{\max}^{(w)} \Sigma_{j \in [d_1]} \left|p_j^{(x)}\right|)\right) =$$
$$O(\log(p_{\max}^{(w)} \Sigma_{j \in [d_1]} \left|p_j^{(x)}\right|)) = O(\log(p_{\max}^{(w)}) + \log(\Sigma_{j \in [d_1]} \left|p_j^{(x)}\right|)) = O(\Sigma_{j \in [d_1]} \log(\left|p_j^{(x)}\right|)) = O(\lambda_{\mathsf{f}}(x))$$
$$\text{(A.2)}$$

As $\lambda_{\mathsf{f}}(\mathrm{ReLU}(w_1 x + b_1)) = O\left(\Sigma_{i \in [d_2]}(\lambda_{\mathsf{f}}(p_i^{(r)}) + \lambda_{\mathsf{f}}(q_i^{(r)}))\right) = O\left(d_2(\max_{i \in [d_2]}(\lambda_{\mathsf{f}}(p_i^{(r)})) + \max_{i \in [d_2]}(\lambda_{\mathsf{f}}(q_i^{(r)})))\right)$, by Equations (A.1) and (A.2) we have $\lambda_{\mathsf{f}}(\mathrm{ReLU}(w_1 x + b_1)) = O(\lambda_{\mathsf{f}}(x))$.

2. Define $q_c^F := \mathrm{cd}(\{\{w_i, b_i : i \in [m]\}\})$ the common denominator of all the parameters in $F$. Let $f : \mathbb{N} \to \mathbb{N}, f(x) = O(\log x)$, let $n \in \mathbb{N}$ and let $M \in \left(\!\!\binom{\mathbb{Q}_k^d}{n}\!\!\right)$ such that $\lambda_{\mathsf{f}}(\mathrm{cd}(M)) = f(n)$. Define $q_c^{F,M} := \mathrm{cd}(\{\{F(x)\}\}_{x \in M})$ then clearly $q_c^{F,M} \le q_c^F \cdot \mathrm{cd}(M)$. Hence, $\lambda_{\mathsf{f}}(q_c^{F,M}) \le \lambda_{\mathsf{f}}(q_c^F \cdot \mathrm{cd}(M)) = O(\log n)$, hence $S_F^{\mathrm{cd}}(n, k, f(n)) = O(\log n)$. $\qquad \square$

### A.1 An illustration of the example in proof of Lemma 3.9

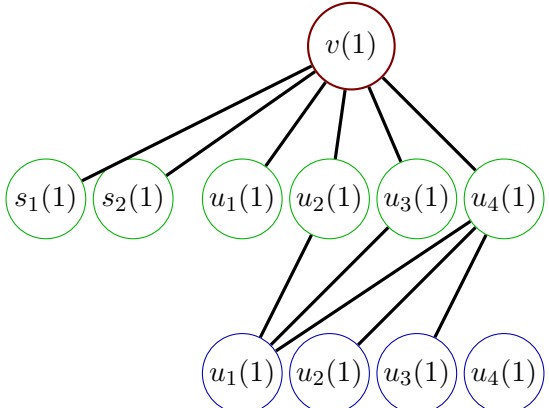

The above depicts the graph $G_K, K \in \mathcal{K}_4, K = \{1, 2, 0, 1, 0\}$. One $u$ vertex is connected to zero $w$ vertices, two are connected to one, zero connected to two, and one $u$ vertex is connected to three $w$ vertices. The initial feature of all vertices is 1. The $\mathrm{CR}^{(2)}$ value of $v$ is

$$\mathsf{cr}_{G_K}^{(2)}(v) = (\{\{1, 1, 1, 1, 1, 1\}\}, \{\{(1, \{\{1\}\}), (1, \{\{1\}\}), (1, \{\{1\}\}), (1, \{\{1, 1\}\}), (1, \{\{1, 1\}\}), (1, \{\{1, 1, 1, 1\}\})\}\})$$

