# OpenReview forum: "Lost in Aggregation: On a Fundamental Expressivity Limit of Message-Passing Graph Neural Networks"
_TMLR — Under review for TMLR_

### Review · Reviewer_KB4u · 2026-04-15

**Summary Of Contributions:**

The paper aims to refine previous analyses of the limits of expressivity of non-recurrent MPGNNs. In particlar, where the Xu et all GIN paper studied non-uniform expressivity (in the sense that for any maximum graph size , a fixed weight GNN can be found that matches 1-WL/CR exactly, and that this is a hard bound in expressive power), this paper studies the decay when using a fixed size network and then going beyond the originally considered graph size. I.e., the paper is a step towards limits in size generalization.

The authors define function characterizations/complexity measures in terms of the bit length of rational vectors serving as the node features/colors to characterize the aggregation function of the MPGNN framework and define "logarithmic" and "sublinear" aggregations, where logarithmic covers the comon max, sum,avg etc. , and sublinear is not instantiated,but used to emphasize the gap between CR2 (running CR for only 2 iterations) and the expressive power of a theoretically more expressive aggregtation method. The authors then show that

*   CR-2 (i.e. running for 2 iterations) can distinguish  $\Omega(c^n)$  graphs where \$n\$ is number of nodes

*   MPGNNs can distinguish  $O(c^{f(n)})$  where \$f(n)\$ is the complexity measure mentiond above

*   so, as n grows larger, the gap grows towards infinity/the fraction of graphs distinguishable graphs over those distinguishable by CR-2 vanishes

*   there are  $\Theta(c^{n^2})$  non isomoprhic graphs (the paper says "doubly exponential" which is a minor error, but the argument works the same), so any aggregation function with bit complexity less than  $f(n)=O(n^2)$  complexity cannot match the expressivtiy required without furhter structural assumptions

This proof strategy (define the information theoretic complexity measure, bound the bit width achievable via induction over layers, convert to a possible equivalence class count) is a useful and to my eyes novel lens on a well established problem

**Additional Comments:**

The core proof architecture is sound and the main theorem is correct. I'd be happy to recommend acceptance after a revision addressing items 1-4. The paper would benefit from being positioned not as uncovering a previously unknown limitation, but as providing a quantitative bit-complexity perspective on a qualitatively known phenomenon, which is still a worthwhile contribution.

**Audience:**

Yes

**Audience Explanation:**

Yes. the bit-complexity framing is a useful addition to the expressivity toolkit and complements existing logical (Barcelo et al., Khalife) and circuit-complexity (Grohe 2023) characterizations. the quantitative counting bound (poly(n) classes) and the CR(2) ratio-to-zero result are new as formal statements even though the qualitative conclusion (fixed MPGNNs are limited uniformly) was already known from the work cited above. with proper positioning this would be a solid contribution.

**Claims And Evidence:**

Yes

**Claims Explanation:**

I put Yes, but this is conditional on addressing the following nitpicky but (I think) clear technical errors

### major:

*   as mentioned, "doubly exponential" is wrong, this doesn't change the asymptotic argument, but changes the rate and gap

*   the definitions for logarithmic and sublinear implicitly assume "input reachable" values as derived by specific features, as stated, you can give a counter example of e.g. choosing  $q_i=1/p_i$  with  $p_i$  being distinct primes larger than \$n\$ , then I think  $\lambda(S_n)$  is something like  $\Theta(n \log n)$  .

*   the paper should define it's assumed bit encoding, I think the implied encoding is magnitude encoding or separately tracking numerators and denominators (where the proofs go through) , but for e.g. reduced form encoding the log identity would fail

*   there is uncited work by S. khalife (2023,2024) on uniform expressivity , and on bottlenecks and oversquash (Alon & yahav 2021, topping et al 2022, di giovanni et all 2023):

    *   Khalife, S. (2023). "The logic of rational graph neural networks." arXiv:2310.13139, proves GNNs with rational activations cannot uniformly express GC2 queries even on depth-2 trees. https\://arxiv.org/abs/2310.13139
    *   Khalife, S. and Basu, A. (2023). "On the power of graph neural networks and the role of the activation function." arXiv:2307.04661, extends the above to piecewise polynomial activations. https\://arxiv.org/abs/2307.04661
    *   Khalife, S. (2024). "Is uniform expressivity too restrictive? Towards efficient expressivity of GNNs." arXiv:2410.01910, shows sigmoid/tanh GNNs can't uniformly express GC2 but can do so with params logarithmic in max degree. https\://arxiv.org/abs/2410.01910
    *   Alon, U. and Yahav, E. (2021). "On the Bottleneck of Graph Neural Networks and its Practical Implications." ICLR 2021, identifies the core over-squashing phenomenon. https\://arxiv.org/abs/2006.05205
    *   Topping, J. et al. (2022). "Understanding over-squashing and bottlenecks on graphs via curvature." ICLR 2022. https\://arxiv.org/abs/2111.14522
    *   Di Giovanni, F. et al. (2023). "On over-squashing in message passing neural networks: The impact of width, depth, and topology." ICML 2023. https\://arxiv.org/abs/2302.02941

### minor:

*   the "two level star" family has counter examples, e.g. n=3,  $K = (0, 3, 0, 0)$ , where each of the three u -vertices is connected to exactly one w -vertex. Then   $\text{dist}(w_1, w_2) = 4$   via the path  $w_1 \to u_1 \to v \to u_2 \to w_2$ . , I think you can use threshold graphs to construct a different family by adding a center adjacent to all nodes in a given threshold graph, and a leaf adjacent only to c. ther’es  $2^{n-1}$  threshold graphs which should give you what you want

*   the CR recurrence on page 5 is circular, the displayed formula defines  $\text{cr}^{(t)}_G(v)$  in terms of  $\text{cr}^{(t)}_G(v)$  and  $\text{cr}^{(t)}_G(w)$  on the right-hand side, when it should be  $t-1$ . clearly a typo, the rest of the paper uses it correctly

*   the diameter definition on page 4 says "the longest simple path between any two of its vertices" which is not the standard meaning of diameter (maximum shortest-path distance). the intended meaning is clear from context

*   the information trace  $I_N(G,v)$  on page 7 omits the message function, it writes  $\text{agg}_i\{\!\{w^{(i-1)}\}\!\}$  instead of  $\text{agg}_i\{\!\{\text{msg}_i(v^{(i-1)}, w^{(i-1)})\}\!\}$ . doesn't affect the proofs since Lemma 3.5 tracks message complexity separately, but the definition should match the computation it's describing

*   the counting formula in Lemma 3.8 states  $|\mathcal{K}_n| = \binom{2n-1}{n-1}$  but I think stars-and-bars for  $n+1$  bins summing to \$n\$ gives  $\binom{2n}{n}$

**Requested Changes:**

### critical

1.  fix or clarify the aggregation class definitions (Definition 3.1 / Example 3.2), either weaken to reachable-input, or state the theorem conditionally, or add a bounded-denominator lemma showing the issue doesn't arise for standard architectures

2.  cite and discuss Khalife (2023, 2024), Khalife & Basu (2023), and the over-squashing literature (Alon & Yahav 2021, Topping et al. 2022, Di Giovanni et al. 2023). clarify what the bit-complexity perspective adds beyond these

3.  fix the "doubly-exponential" terminology (the standard asymptotic is  $2^{\Theta(n^2)}$ , see Harary & Palmer, *Graphical Enumeration*, 1973, or Flajolet & Sedgewick, *Analytic Combinatorics*, 2009, p. 106)

4.  state the rational encoding convention explicitly

### non critical

the other fixes/caveats above

---

> ### Author Response · Authors · 2026-05-11
> **response to review**
>
> We thank the reviewer for the thorough and useful review. Below is our detailed response.
>
> In Summary of Contributions:
> - "originally considered graph size". To clarify: there is no "originally considered". If the reviewer means "training-graphs size" then ok.
> - "there are $\Theta(c^{n^2})$”. To clarify: This is not a result, we mention it to put things in perspective for an unfamiliar reader. MPGNNs are limited far beyond that - by CR - and regardless of aggregation.
>
> In Major:
> - “"doubly exponential"”. we will correct that.
> - “input reachable”. Meaning not clear. If the reviewer means “input domain” then the definitions only say “if...then...”, they do not assume anything. The proof of Lemma 5.1 does make use of the initial features and it is true that if the features are linear in the graph size then the proof does not apply as is. We will emphasizing is.
> - Explicit encoding.
> 1. Reduced form is a specific case of fractional representation.
> 2. We agree that the encoding, specifically fractional representation, has to be taken into account. In fact, after doing that we find that the example no longer works as is. We will revise the definition of logarithmic aggregation so that it captures a large class of aggregations, including sum and max for example, and the results still apply to it. Note however that the results will probably not apply to average aggregation, and we might be able show that average aggregation can distinguish asymptotically more graphs.
> - Citing of “"The logic of rational..." The paper considers activations less powerful than ReLU. We focus on limitations rooted in the aggregation function. Hence, results about activation-rooted limitations are not immediately relevant.
> - Citing of “the role of the activation function”. We will cite it as it considers powerful activations which we mention.
> - Citing of “uniform expressivity too restrictive?...”. It concerns non-uniform expressivity, which is not our focus. However, we will consider citing it.
> - Citing of over-squashing papers.
> To clarify: The paper does not directly address a specific practical “well-known problem”. As for over-squashing in particular, all papers mentioned by the author state explicitly, many dozens of times, that over-squashing concern long-range interaction. As stated in our concluding remarks, the direct implication to practice of our results is when the target function determines a polynomial (or sub-exponential) number of equivalence classes. This is a different description than the statement of the over-squashing problem. The limitation we describe is relevant already for range 2 - not much of a “long range”, and on the other hand the over-squashing problem is not necessarily a matter of a fine-grain target function. Although both relate to aggregation as the culprit, it plays a different role in each. In addition, we characterize a very general class of aggregations, while over-squashing works usually focus on common specific ones.
> To summarize: The aggregation class scope, the problem definition, and the proven (or hypothesized in the over-squashing case) roles that the aggregation plays in it, are different between our work and works about over-squashing, hence we do not cite them as related work.
>
> In Minor:
> - “the two level star...”.
> 1. for $K = (0, 3, 0, 0)$ each of the 3 u vertices are connected to w_1, see the precise definition of E(G).
> 2. "counter example" is unclear, counter to what? we describe a family of graphs and prove that it has a certain size and that all of the graphs in it are distinguishable by CR^2. 3. The definition indeed allows (by mistake) a diameter > 2. It does not matter to the main point, but is inconsistent with "diameter-2" statements. We fix it by adding a special w_0 vertex that is connected to v and all u.
> - "CR recurrence on page 5...”. we will fix it.
> - diameter definition. right, we mean “longest shortest...”, we will fix it.
> - “$I_N(G,v)$ on page 7”. we will fix it.
> - “counting formula in Lemma 3.8”. we will fix it.
>
> Others:
> “MPGNNs are limited uniformly was already known”
> “positioned not as uncovering a previously unknown limitation”
> We beg to differ on the reviewer’s framing of results. To use his/her terminology, almost every complexity result in CS is essentially “a perspective on qualitatively known phenomenon”. “MPGNNs are limited”, even limitation by 1-WL/CR, is very general. For anything stronger, our work indeed uncovers unknown limitations, in several ways:
> 1. We prove a precise limitation in terms of number of induced equivalence classes.
> 2. We characterize a very broad class of computable aggregations. Prior works, including those mentioned by the reviewer, that account for computable aggregations consider only specific ones.
> 3. Relating to the reviewer’s terminology and insinuated importance scale, the CR$^{(2)}$ limitation that we show is a “qualitative phenomenon”, stronger than the known CR limitation, and in a sense tight.

---

### Review · Reviewer_bG94 · 2026-05-10

**Summary Of Contributions:**

This paper studies the uniform distinguishing power of message-passing graph neural networks (MP-GNNs) under a broad class of aggregation functions. The authors introduce an output-size complexity notion for aggregation functions, distinguishing logarithmic and sublinear aggregations. They argue that common aggregations such as sum, mean, and max fall into the logarithmic class. The main result is that any fixed MP-GNN using logarithmic aggregations induces only polynomially many equivalence classes over graphs of size n, while MP-GNNs using sublinear aggregations induce only sub-exponentially many equivalence classes. The result is established both for node-level distinguishability and graph-level readout settings.

**Audience:**

Yes

**Audience Explanation:**

The paper addresses a fundamental question about the expressivity of MP-GNNs, which remains a central topic in graph representation learning. The distinction between non-uniform and uniform distinguishing power is particularly relevant: many classical expressivity results show that MP-GNNs can match Color Refinement in a non-uniform sense, but practical learned models are fixed architectures and fixed parameterizations deployed across graphs of varying sizes. The paper’s focus on the uniform setting therefore addresses a meaningful gap between theory and practice.

The result is also relevant to researchers studying graph learning for molecules, knowledge graphs, combinatorial optimization, and other structured domains, where generalization across graph sizes is important. Even if the result is mainly theoretical, it provides a useful lens for understanding when standard MP-GNNs with common aggregations may fundamentally fail to distinguish graph structures.

That said, the paper is currently most likely to appeal to a theoretical subset of the TMLR audience. To broaden its impact, the authors should improve the motivation, provide more concrete examples, and clarify the implications for commonly used GNN architectures.

**Broader Impact Concerns:**

I do not see significant negative broader impact concerns.

**Claims And Evidence:**

Yes

**Claims Explanation:**

The main claims are mostly supported at a high level, but several aspects of the presentation and some technical details need clarification before the evidence can be considered fully convincing.

The central proof strategy is elegant: the paper defines an information-complexity measure \(L_N(n)\), argues that the number of distinguishable node or graph classes is upper bounded by \(2^{L_N(n)}\), and then proves that \(L_N(n)=O(\log n)\) for logarithmic aggregations and \(L_N(n)=o(n)\) for sublinear aggregations. This gives the stated polynomial and sub-exponential upper bounds. The proof structure is clear and appears plausible.

However, I have several concerns about clarity and rigor. First, the paper repeatedly says that the number of non-isomorphic graphs is “doubly-exponential” in the number of vertices. In the standard asymptotic sense, the number of unlabeled graphs on \(n\) vertices is approximately \(2^{\Theta(n^2)}\), which is super-exponential in \(n\), but not doubly exponential in the usual meaning \(2^{2^{\Theta(n)}}\). This should be corrected throughout the paper, since it affects how readers interpret the quantitative gap.

Second, the definition of the aggregation class is important but somewhat nonstandard. The authors should provide more intuition and examples beyond sum/mean/max. The claim that the class captures “most conceivable aggregations” is too broad as written. It would be better to say that it captures many commonly used finite-dimensional aggregations, and then explicitly discuss where attention, softmax, PNA-style aggregations, histogram/binning, sorting/top-\(k\), or quantile-type aggregations fall.

Third, the paper’s practical implication needs to be stated more carefully. The formal results concern distinguishability under rational, finite-bit representations and fixed-depth, fixed-parameter MP-GNNs. This is meaningful, but the connection to practical approximation of target functions should be more carefully qualified. For example, the conclusion that a target function with more than polynomially many values cannot be represented by logarithmic-aggregation MP-GNNs is correct only insofar as the target requires distinguishing all those equivalence classes within the considered graph domain.

Fourth, the comparison with CR(2) is interesting and useful, but it would benefit from more explanation. The construction in Lemma 3.8 is not difficult, but the reader has to work to see why the CR(2) colors are all distinct across the constructed graphs. A short illustrative example for small \(n\) would make the argument much clearer.

Finally, there are several typographical and notation issues that reduce confidence in the manuscript. For example, the title contains “Fundamantal” instead of “Fundamental,” and there are several minor spelling issues such as “straghitforward” and “copmlexity.” The notation around \(I_N(G,v)\), \(L_N(n)\), and the aggregation outputs should also be made more consistent.

**Requested Changes:**

1. **Correct the “doubly-exponential” terminology.**
   The number of non-isomorphic graphs on \(n\) vertices should be described as \(2^{\Theta(n^2)}\), or super-exponential in \(n\), rather than doubly exponential in the usual sense. This appears in the abstract, main results discussion, and conclusion.

2. **Clarify the scope of the aggregation class.**
   The phrase “most conceivable aggregations” is too strong. Please provide a more precise characterization and include a table or discussion covering common aggregations such as sum, mean, max, min, degree-normalized sum, PNA-style aggregations, attention/softmax, top-\(k\), sorting-based aggregations, histogram aggregations, and quantile aggregations.

3. **Improve the intuition behind Definition 3.1.**
   The definitions of logarithmic and sublinear aggregation are central to the paper, but they are not immediately intuitive. Please add a paragraph explaining why output bit-length is the right complexity measure, how it relates to finite precision computation, and why the dependence on both multiset size \(n\) and input bit-length \(k\) is necessary.

4. **Strengthen the explanation of Lemma 3.3.**
   The claim that the aggregation-bit sequence \(I_N(G,v)\) determines the final MP-GNN output should be explained more carefully. In particular, the definition of \(I_N(G,v)\) should consistently include the actual aggregation outputs applied to the message values, and the proof should explicitly state the deterministic induction over layers.

5. **Clarify graph-level readout arguments.**
   The graph-level part of Theorem 3.7 is described as “relatively straightforward.” Since graph-level distinguishability is one of the paper’s advertised contributions, the proof should be written out in more detail rather than deferred to a brief adaptation.

6. **Provide a small example for the CR(2) lower-bound construction.**
   The two-level star construction in Lemma 3.8 is important. A small example, perhaps for \(n=3\), would help readers understand why different choices of the degree profile lead to different CR(2) colors.

7. **Tone down or qualify the practical conclusion.**
   The conclusion should distinguish between formal distinguishability limits and practical learning performance. The paper should avoid implying that all target functions with many possible values are practically impossible unless the target indeed requires separating all corresponding graph equivalence classes.

8. **Fix typos and notation issues.**
   Examples include “Fundamantal” in the title, “straghitforward,” “copmlexity,” and several inconsistent or malformed equations due to formatting. The manuscript would benefit from a careful proofreading pass.

---

> ### Author Response · Authors · 2026-06-01
> **response to review**
>
> We thank the reviewer for the thorough and useful review. Below is our detailed response.
>
> In Requested Changes:
> - “doubly-exponential...”. We will fix that.
> - "scope of the aggregation class..." 1. The definition will be changed to accommodate for the rational representation. 2. We already discuss the fact that aggregations that use softmax are not necessarily logarithmic. 3. With the new definition also avg is not guaranteed to be logarithmic - but maybe we still provide a useful upper bound for it. 4. We will add theorem that shows how the definition captures aggregations that comprise a useful range of operations, and a corollary that shows how common aggregations (additional to sum;max) such as binning; k-selection; quantile-selection; and concatenation of multiple logarithmic aggregations; are logarithmic.
> - "Improve the intuition behind Definition 3.1..." As its introductory sentence says, Lemma 3.3 is the justification for the complexity measure we define. The properties of the input multiset i.e. element-count and element bit-length, are the two quantities that may affect the bit-complexity of the aggregation. In theory, for example, a node's feature bit-length may be linear in the graph size, in which case the bit-length of even a max-aggregation will be linear no-matter the multiset size (i.e. number of neighbors).
> - "Strengthen the explanation of Lemma 3.3...". We will fix the definition to include the message function. We will detail the induction.
> - "Clarify graph-level readout arguments..." We will give more details.
> - "Provide a small example for the CR(2)..." We will explain further why the colors are distinct,      possibly adding an illustration of an example graph.
> -  "Tone down or qualify the practical conclusion..." We write that when the target function
>     "ASSUMES a greater-than polynomial number of values then..." - rather than that its theoretical range has these many values, meaning that indeed it requires distinguishing these many equivalence classes within the considered graph domain. We will try to make the meaning clearer.
> -  Fix typos and notation issues. We will fix the typos. We are not sure to which inconsistencies the reviewer refers to, we will be happy to know and address them.

---

### Review · Reviewer_UbqD · 2026-05-27

**Summary Of Contributions:**

The paper considers the _node_-level and _graph_-level distinguishing-power of message-passing graph neural networks with rational weights and logarithmic, and more generally sublinear, aggregation functions (referred to as MP-GNNs). The power of MP-GNNs is then compared to two iterations of the 1-dimensional Weisfeiler-Leman color refinement algorithm (1-WL). The main result is that MP-GNNs with an arbitrary number of iterations can only distinguish a polynomial (for logarithmic aggregation functions; subexponential for sublinear aggregation functions) number of node colors, while for 1-WL, there is at least one graph family for which it can distinguish exponentially many node colors after only two iterations.

Strengths:

* The paper provides a more precise characterization of GNN limitations in comparison to previous work by analyzing the granularity of induced node color equivalence classes.
* Focusing on rational-weight GNNs and commonly used aggregation functions makes the results more relevant to practical architectures.

Weaknesses:

* The theoretical results are proven for node color equivalence classes, and not graph-level equivalence classes. This undermines the motivation and objective of the paper, especially as it compares with color refinement algorithms which serve as heuristic for graph isomorphism testing.
* The clarity of the paper has room for improvement; it contains several typos, inconsistent notation and insufficient pointers to references. Furtermore, the writing often relies on unsupported assertions and speculative claims.
* There is no substantive comparison with related work, e.g., differences in assumptions and scope are not discussed in more detail.

**Additional Comments:**

I have several questions concerning notation and definitions, which I list below. It is possible that resolving some of them might lead to follow-up questions about the soundness of the results.

* In the paper, the _diameter_ of a graph is defined as the _longest simple path_ between any two vertices. The paper states that a sum-aggregation MP-GNN matches CR$^{(1)}$ for diameter-1 graphs (point 2. in page 2). Given the definition, diameter-1 graphs comprise any collection of single edges; is this the intended meaning? Or is the diameter erroneously defined and meant in the more standard definition as the longest _shortest_ path between any two vertices, in which case diameter-1 graphs would be all complete graphs? In either case, it would be helpful if you could elaborate on this observation.
* The paper motivates their work by the need of a more fine-grained characterization of the distinguishing-power of MP-GNNs than [CR$^{(1)}$, CR]. What is the meaning of this interval, i.e., the distinguishing power between one iteration of CR and CR more generally?
* In paragraph **New Results** it is stated that the number of non-isomorphic graphs is doubly-exponential; to the best of my knowledge it is $2^{O(n^2)}$, thus single-exponential. Could you elaborate how you arrived at this bound?
* Are there any assumptions on the sets $S$ and $T$ for featured graphs? Are they finite/bounded? What is the difference between $S$ and $T$?

Minor remarks:

* There are several typos throughout the paper, e.g., ''Fundamantal'' in the title, ''may required'' in the abstract.
* Notation: $k$ is used inconsistently; as a scalar in the preliminaries and as a function in Example 3.2; similarly, $x_i$ appears without definition; $q$ is undefined in the last sentence of paragraph **Featured Graph**.
* The color refinement update step is incorrect; it should be $(t+1)$ on the left-hand side, or $(t-1)$ on the right-hand side.
* Footnote 1 appears on the wrong page.
* Consider using \citep and \citet for citations consistently to improve readability.

**Audience:**

Yes

**Audience Explanation:**

Overall, analyzing the expressive power of graph neural networks via their ability to induce equivalence classes for graphs is a very interesting direction; looking at rational-weight GNNs specifically is also relatively under-explored and has been mainly considered from a logic perspective so far [1,2].

[1] Martin Grohe: The Descriptive Complexity of Graph Neural Networks, TheoretiCS, 2024.

[2] Veeti Ahvonen, Damian Heiman, Antti Kuusisto, Carsten Lutz: Logical Characterizations of Recurrent Graph Neural Networks with Reals and Floats, NeurIPS, 2024.

**Claims And Evidence:**

No

**Claims Explanation:**

The main theoretical contribution, to my understanding, is that there exists at least one graph family for which the node-distinguishing power of a specific family of message passing graph neural networks induces a less fine partition on the node colors than the 1-dimensional Weisfeiler Leman algorithm does after two iterations. For expressivity results in graph machine learning, one is usually interested in how good an algorithm/machine learning model is in distinguishing between non-isomorphic graphs. The paper states that it is ''relatively straightforward'' (e.g., in the proofs of Lemma 3.3, Theorem 3.7 and Lemma 3.8) to adapt their setting to distinguishing graphs; however, there is no formal proof nor is this further discussed. For instance, it is currently unclear to me how the exponential lower bound for _node_ color equivalence classes of CR can be transformed into a bound for _graph_ equivalence classes.

**Requested Changes:**

Required for acceptance:

* Address the missing link between node-distinguishing power and graph-distinguishing power.
* Clarify the practical significance of the polynomial node equivalence class bound. In particular, the two-level star graphs used to establish the exponential lower bound for CR$^{(2)}$ are never shown to be indistinguishable by MP-GNNs. To my current understanding, Corollary 3.9 places two unrelated bounds next to each other rather than identifying a concrete family of graphs that separates the two classes.
* Write a proper related work section that situates the results relative to prior work on GNN expressivity and WL, with precise discussion of how assumptions and conclusions differ.
* Clearly state the paper's contributions and their relationship to existing work, and ensure all claims are supported by proofs or pointers to the literature.
* Improve writing quality throughout: fix typos, clean up notation, and ensure consistency.

Suggested (would strengthen the work):

* Rewrite the introduction to provide sufficient motivation and discuss related work on aggregation functions and the expressive power of GNNs.
* Remark 3.6: Consider formally extending the results to a broader range of MP-GNN architectures.
* Identify specific failure cases for rational-weight GNNs to make the limitations more concrete. E.g., could you give a specific example of two graphs that are distinguished within two iterations of CR, but not with MP-GNNs?

---

> ### Author Response · Authors · 2026-06-05
> **response to review**
>
> We thank the reviewer for the thorough and useful review. Below is our detailed response.
>
> In Weaknesses:
> - "There is no substantive comparison..." The introduction describes related results and the important differences in settings. These are embedded in the whole description of the motivation and current landscape, rather than as a concentrated list under a 'Related Work' title. This is a choice we intend to keep.
> - "The main theoretical contribution..." As we explicitly state, in the abstract; introduction; main results; and concluding remarks, 1. Our contribution is first and foremost the upper-bounding of the distinguishing power of a general class of aggregations by a significant bound - polynomial number of equivalence classes vs the super-exponential number of non-isomorphic graphs. 2. The measure and comparison with Color Refinement is a reference point that we add, to examine the first result through another familiar and meaningful perspective.
> - "it is currently unclear how exponential lower bound..." The CR value of a graph is the multiset of the CR values of its vertices. We show that the CR value of the center-vertex of our star graph is unique for each of those graphs - across all our graphs and vertices, hence necessarily the multiset of vertices values is unique for each of those graphs - it differs from the others by containing the unique center-vertex value. We will add these details (in a formal way) to the paper.
>
> In Requested Changes:
> - "Address the missing link..." Responded above.
> - "Clarify the practical significance..." We explicitly state the importance of distinguishing power in the introduction, and the practical implication of number of equivalence classes in the concluding remarks.
> - "The two-level star graphs used to establish the exponential lower bound for CR are never shown..." Yes, Theorem 3.7 directly implies that no specific MP-GNN model with logarithmic aggregations can distinguish all our star graphs: By the theorem, for each model there exists a size (2n+1) such that the number of star graphs of size (2n+1) is greater than the number of different values that the model can produce. Hence, necessarily there exist two graphs that are assigned the same value by the model i.e. they are indistinguishable by the model. Also, as the family of star graphs is distinguishable by CR$^{(2)}$, it indeed separates logarithmic-aggregation MP-GNN models from CR$^{(2)}$.
> Note that we examine the ability of a model to distinguish a whole domain of graphs. If the requirement is to distinguish merely one specific pair of graphs then for each pair (distinguishable by CR) there may be a model that distinguishes that pair. However, such a weak requirement is not relevant to practice - we want to learn models that will infer on a domain of graphs, not on one specific pair.
> - "Write a proper related work section..." Responded above.
> - "Clearly state the paper's contributions..." The paper's contribution is clearly stated in the introduction's 'New Results' section, after a detailed survey of related literature, with relevant references.
> - "Fix typos..." We will fix typos.
>
> In Suggested:
> - "Remark 3.6: Consider formally extending..." This is open for future work.
>
> In Additional Comments:
> - "the diameter of a graph is defined..." 1. This is an error in the definition, we mean the definition of longest shortest path, we will fix it. 2. The restriction to "diameter-1" is unnecessary and reduces significance. A trivial sum MP-GNN subsumes CR$^{(1)}$ on all graphs, since CR$^{(1)}$ is essentially a neighbor-count. We may drop also other diameter restrictions.
> - "The paper motivates their work by..." The interval is number of CR iterations, from 1 and up to graph size.
> - "it is stated that... is doubly-exponential..." This is inaccurate by us, the number is at least $2^{O(n^2)}$, i.e. super-exponential rather than doubly-exponential. We will fix it.
> - "Are there any assumptions on the sets..." In the general definition of a featured graph and a featured transformation, S and T are merely symbols for any sets that are the input-features domain and the output-features domain. The specific domain that we consider in the paper is defined explicitly.
>
> In Minor remarks:
> - "There are several typos..." We will fix them.
> - "k is used inconsistently..." A symbol may be used in different meanings in separate scopes. This is not inconsistent.
> - "x_i appears without definition..." Where?
> - "q is undefined..." d is undefined, q is defined and not used. This is a typo we will fix.
> - "The color refinement update step is incorrect..." This is a typo we will fix.
> - "Footnote 1 appears..." We will fix that.
> - "Consider using \citep..." We will consider using \citep.

---

### Author Response · Authors · 2026-06-15
**new revision**

Dear all,

We have uploaded a new and much-improved revision, fixing many typos and small mistakes/inaccuracies; improving design and layout; and adding explanations to improve readability, which we consider very close to the provisional final revision. Note that it does not include a specific example of a $G_K$ graph and the corresponding CR$^{(2)}$ values, which was requested by reviewer bG9410. We intend to add such an example for the provisional final revision.

Thank you for your time. Kind regards,

---

### Author Response · Authors · 2026-06-22
**Provisional final revision**

Dear all,

We have uploaded a provisional final revision. Besides small fixes, clarifications, and additional information regarding mean-aggregation distinguishing-power upper bound, it includes an example (and illustration in the appendix) of a specific $G_K$ graph, which was requested by reviewer bG9410.

We apologize for the delay, and appreciate your understanding. We look forward to your comments and decisions.

Thank you for your time. Kind regards,